# Two-Stage Coverage Expansion for Cross-Domain Offline Reinforcement Learning via Score-Based Generative Modeling

## Abstract

Cross-domain reinforcement learning (RL) aims to transfer knowledge from a source domain to a target domain with different dynamics, but existing approaches often directly reuse source transitions, which can lead to severe distributional mismatch and performance degradation when the domain gap is large or target data is scarce. We propose Two-stage Coverage Expansion (TCE), a dual score-based generative framework that first expands state coverage through a mixture-based state score network and then aligns transitions with target-domain dynamics using a target-transition score network. This two-stage design broadens the effective support of the target dataset while mitigating harmful distributional shift, enabling more improved policy learning under limited target data. Extensive experiments on diverse cross-domain benchmarks demonstrate that TCE consistently outperforms state-of-the-art cross-domain RL baselines, achieving substantial gains even under large domain gaps and extremely small target datasets.

## 1 Introduction

Cross-domain reinforcement learning (cross-domain RL) fundamentally aims to adapt or transfer a learned policy from a source domain to a target domain with potentially different environment dynamics. This problem setting frequently arises in real-world applications such as controlling heterogeneous robots, simulation-to-real autonomous driving, and medical decision making (Gottesman et al., 2018; Yurtsever et al., 2020). To address such cross-domain scenarios, various RL-based methods have been proposed (Eysenbach et al., 2020; Kim et al., 2020). However, most existing methods assume that online interaction with either the source or target domain is feasible, thereby allowing the data collection during training. In practice, this assumption rarely holds. In many realistic cross-domain applications, online interaction is severely restricted due to cost and safety concerns, and in some cases, it is entirely infeasible (Levine et al., 2020). Consequently, cross-domain offline RL, where only pre-collected datasets from both domains are available, has become an important research direction for enabling cost-efficient learning without online interaction (Liu et al., 2022).

Early studies on cross-domain offline RL mainly focused on selecting source-domain data similar to the target domain or applying mutual-information-based filtering (Poole et al., 2019; Guo et al., 2022), implicitly assuming that sufficient target data is available (Xu et al., 2023; Lyu et al., 2024a). When the target dataset is abundant, however, single-domain offline RL algorithms such as CQL (Kumar et al., 2020), IQL (Kostrikov et al., 2022), and ReBRAC (Wu et al., 2019) already perform strongly, often making additional source data unnecessary or even harmful. More recent work has therefore explored settings where the amount of target data is extremely limited (Wen et al., 2024; Lyu et al., 2025). Nevertheless, we find that when the domain gap between the source and target is large, simply incorporating source transitions can still introduce severe distributional mismatch and may degrade performance rather than improve it.

To address these challenges, we propose *Two-stage Coverage Expansion (TCE)*, a dual score-based generative framework with stochastic differential equations (SDEs) that, rather than merely selecting source data, constructs a mixture distribution with target-like transitions to broaden target coverage and reduce distributional mismatch, supported by concrete theoretical analysis. TCE consists of 1) a *mixture-based state score network* trained on a controllable mixture of source and target states to appropriately broaden the target state space, and 2) a *target-transition score network* trained only on

target transitions to produce state transitions consistent with target dynamics. At inference time, TCE performs two-stage sampling: first drawing diverse states from the state score network using the SDE sampler, and then generating target-like next states from the transition score network conditioned on the sampled states. Using auxiliary models and Z-score–based filtering, TCE constructs a high-quality augmented dataset that increases target-domain transition coverage while minimizing distributional mismatch. This principled two-stage design is, to our knowledge, the first cross-domain offline RL approach to jointly control state coverage expansion and align transitions with target dynamics. Across diverse cross-domain environments, TCE shows substantial performance gains over state-of-the-art cross-domain offline RL baselines.

## 2 RELATED WORKS

**Cross-Domain Reinforcement Learning.** Early cross-domain RL methods rely on online data collection and focus on domain-invariant representations or adversarial domain alignment to facilitate transfer (Eysenbach et al., 2020; Yu et al., 2021). Cross-domain imitation learning extends this by leveraging demonstrations across domains to generalize behavior without explicit rewards (Kim et al., 2020; Fickinger et al., 2022; Choi et al., 2023). Approaches for domain adaptive imitation learning target robustness against environmental dynamics variations (Chae et al., 2022). However, many assume at least some level of online interaction or sufficient target data (Xu et al., 2023; Lyu et al., 2024a), which limits their use in purely offline settings. Offline cross-domain RL methods address this constraint (Wen et al., 2024; Lyu et al., 2025) but face challenges when the domain gap is large and target data is limited. In addition, a recent study has explored generating target-aligned source data in order to mitigate the domain gap (Le Pham Van et al., 2025).

**Offline Reinforcement Learning.** Offline RL algorithms such as Implicit Q-Learning (IQL) (Kostrikov et al., 2022) and Conservative Q-Learning (CQL) (Kumar et al., 2020) have demonstrated strong single-domain performance on static datasets like D4RL (Fu et al., 2020). Nonetheless, handling multi-domain data and domain shifts remains challenging (Liu et al., 2022; 2024). Filtering strategies leveraging mutual information (Poole et al., 2019; Guo et al., 2022) and behavior regularization (Wu et al., 2019) are used to mitigate distributional shifts, but cross-domain offline learning with limited target data is under-explored. Some studies have proposed selectively incorporating source data similar to the target domain to alleviate this challenge (Wen et al., 2024; Lyu et al., 2025), but when the domain gap is large or selection is suboptimal, these methods may fail to improve or even hinder policy learning. Recently, diffusion-based techniques have shown promise by providing effective data augmentation and model learning strategies in offline RL, further improving policy performance on limited datasets (Li et al., 2024; Luo et al., 2025). In addition, Transformer-based methods that perform offline learning over a distribution of tasks to enable generalization have also been investigated (Wang et al., 2024).

**Score-Based Models and Diffusion Processes.** Two principal approaches to score-based generative modeling have independently advanced high-quality sample generation: denoising score matching, which estimates gradients of data log-density at multiple noise scales (Song & Ermon, 2019), and diffusion models, which progressively corrupt and then denoise data through a series of intermediate steps (Ho et al., 2020). The stochastic differential equations (SDEs) framework provides a unifying view, generalizing both approaches and enabling principled continuous-time sampling procedures (Song et al., 2020). Our method leverages this SDEs formalism to jointly train label-conditioned score models over states and transitions, combined with outlier filtering, facilitating reliable and domain-aligned data generation in cross-domain offline RL.

## 3 BACKGROUND

### 3.1 MARKOV DECISION PROCESS AND CROSS-DOMAIN OFFLINE SETUP

We define a Markov Decision Process (MDP) as $\mathcal{M} = (\mathcal{S}, \mathcal{A}, P_{\mathcal{M}}, r, \gamma)$, where $\mathcal{S}$ is the state space, $\mathcal{A}$ the action space, $P_{\mathcal{M}}$ the transition dynamics, $r$ the reward function, and $\gamma$ the discount factor. In the cross-domain setting, we assume access to a source domain $\mathcal{M}_{\text{src}} = (\mathcal{S}, \mathcal{A}, P_{\text{src}}, R, \gamma)$ and a target domain $\mathcal{M}_{\text{tar}} = (\mathcal{S}, \mathcal{A}, P_{\text{tar}}, R, \gamma)$, which share the same state and action spaces as well as the reward function but differ in their transition dynamics, i.e., $P_{\text{src}} \neq P_{\text{tar}}$. In the cross-domain offline setting, the agent cannot interact with either domain and must rely solely on pre-collected transitions $(s_t, a_t, r_t, s_{t+1})$, where $s_t \in \mathcal{S}$ denotes the state, $a_t \in \mathcal{A}$ the action, $r_t = R(s_t, a_t)$ the reward, and $s_{t+1} \sim P(\cdot|s_t, a_t)$ the next state with $P = P_{\text{src}}$ or $P = P_{\text{tar}}$. We denote the datasets collected

from the source and target domains as $\mathcal{D}_{\text{src}}$ and $\mathcal{D}_{\text{tar}}$, respectively, under the practical constraint that $|\mathcal{D}_{\text{tar}}| \ll |\mathcal{D}_{\text{src}}|$, making direct policy learning on the target domain challenging.

## 3.2 SCORE-BASED GENERATIVE MODELS WITH SDES

Generative models aim to learn the data distribution $p_{\text{data}}(x)$ and generate realistic samples, with representative approaches including Generative Adversarial Networks (GANs) (Goodfellow et al., 2014), Variational Auto-Encoders (VAEs) (Kingma & Welling, 2013), and diffusion models (Ho et al., 2020). Among these, score-based generative models with SDEs (Song et al., 2020) offer a continuous-time formulation of diffusion, support flexible noise scheduling, and enable efficient sampling and likelihood computation via the probability-flow ODE. While conditioning is optional, we explicitly include a condition $c$ so that generation is guided by $c$. Given a clean sample $x^0$ conditioned on $c$, we perturb it with Gaussian noise $x^\tau = x^0 + \sigma(\tau)z$, where $z \sim \mathcal{N}(0, I)$, $\tau \in [0, 1]$ is the continuous noise level, and $\sigma(\tau)$ is noise scale. A score network $q_\theta(x, \tau \mid c)$ is then trained to approximate the conditional score $\nabla_x \log p_\tau(x \mid c)$ via denoising score matching:

$$\mathcal{L}_{\text{score}}(\theta) = \mathbb{E}_{\tau, (x^0, c)}\Big[\lambda(\tau) \parallel q_\theta(x^\tau, \tau \mid c) + z/\sigma(\tau)\parallel_2^2\Big], \tag{1}$$

where $\lambda(\tau)$ is a time-dependent weight. At sampling time, starting from $x^1 \sim \mathcal{N}(0, \sigma(1)^2 I)$, samples are generated by solving the discretized reverse SDE using the Predictor–Corrector sampler with discretized noise levels $\tau^k$ ($1 = \tau^K > \cdots > \tau^0 = 0$):

$$x^{k-1} = x^k + \big[f(x^k, \tau^k) - g(\tau^k)^2 q_\theta(x^k, \tau^k \mid c)\big]\Delta\tau^k + g(\tau^k)\sqrt{\Delta\tau^k}\,\xi^k, \quad \xi^k \sim \mathcal{N}(0, I), \tag{2}$$

where $f$ and $g$ denote the drift and diffusion coefficients of the forward SDE and $\Delta\tau^k = \tau^{k-1} - \tau^k$ is the step size. In this work, we additionally apply a Langevin corrector (Song & Ermon, 2019) after each predictor step to further refine sample quality. The implementation details of $\lambda(\tau)$, $f(x, \tau)$, $g(\tau)$, the step size, and the Langevin corrector are provided in Appendix B.

## 4 METHODOLOGY

### 4.1 MOTIVATION

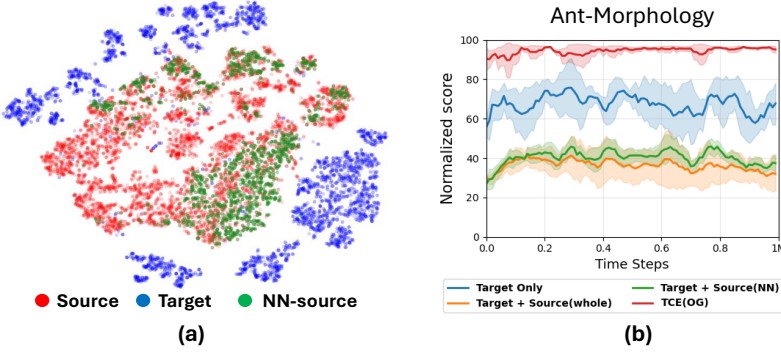

**Figure 1**: (a) t-SNE visualization of state transitions $(s_t, s_{t+1})$ from the source data, target data, and NN-source. The NN-source set is constructed by selecting source samples nearest to the target data, and all datasets are randomly subsampled to have equal size for comparison. (b) Performance comparison of IQL convergence across different datasets: target-only, target with NN-source data, target with the entire source data, and our proposed TCE. Here, 'TCE (OG)' denotes a variant of the proposed TCE framework that relies solely on generated samples and does not use any source data.

Most existing cross-domain offline RL methods address the scarcity of target data by reusing source-domain transitions that are closest to the target data under some distance metric (Wen et al., 2024; Lyu et al., 2025). Although the definition of distance varies, these methods share the assumption that nearby source data always improves target-domain learning. We show that this assumption can harm performance when the domain gap is large. Fig. 1(a) shows a t-SNE visualization of state transitions $(s_t, s_{t+1})$ in the MuJoCo Ant environment, where the source data are collected from an agent with a different morphology from the target. The visualization indicates that the two domains have little

overlap, revealing a significant domain gap. To examine the effect of reusing source data in this setting, we select the subset of source transitions that are nearest neighbors to the target transitions (NN-source in Fig. 1(a)) and train policies with IQL under three datasets: target-only, target plus NN-source, and target plus all source data. As shown in Fig. 1(b), both NN-selected and full-source augmentation result in worse performance than target-only training, suggesting that naive source reuse can hinder learning when the domain gap is large.

To address this issue, we introduce Two-stage Coverage Expansion (TCE) as described in Section 1, a two-stage score-based data augmentation approach that leverages source data to expand target state coverage and generate transitions aligned with target dynamics, thereby reducing distributional mismatch. Fig. 1(b) further shows that augmenting the target data with transitions generated by TCE yields markedly better performance. In contrast to directly reusing source data, either through nearest-neighbor selection or by using the full source dataset, TCE expands the state space with its mixture-based generator and generates target-consistent transitions with the target-transition generator, resulting in improved policy learning. Although limited target data can still cause some overfitting, the generated transitions remain closer to the target domain than direct source reuse, contributing to the observed gains. The next section presents the algorithmic details of TCE.

### 4.2 TWO-STAGE COVERAGE EXPANSION VIA SCORE-BASED GENERATIVE MODELING

To address the limitation identified in the motivation, we propose TCE, which does not simply filter source data but instead expands state coverage by generating target like transitions. Before introducing the full algorithmic details of TCE, we first establish its necessity from a gap bound perspective. To this end, for a given MDP $\mathcal{M}$ and policy $\pi$, let $\rho_{\mathcal{M}}^{\pi}(s,a) := (1-\gamma)\sum_{t=0}^{\infty}\gamma^t P_{\mathcal{M}}(s \mid s_t, a)\pi(a \mid s_t)$ be the discounted occupancy measure, and let $V_{\mathcal{M}}^{\pi}(s_t) = \mathbb{E}_{\rho_{\mathcal{M}}^{\pi}}\left[\sum_{l=t}^{\infty}\gamma^l r(s_l, a_l)\right]$ be the value function. Our objective is to maximize the average return $\eta_{\mathcal{M}}(\pi) = \mathbb{E}_{\rho_{\mathcal{M}}^{\pi}}[r(s,a)]$ using both $\mathcal{D}_{\mathrm{src}}$ and $\mathcal{D}_{\mathrm{tar}}$. Let $P_{\mathrm{src}}$ denote the source transition, and let $\hat{P}_{\mathrm{tar}}$ be an approximate target transition used for coverage expansion. We define their mixture as $P_{\mathrm{mix}} = \lambda P_{\mathrm{src}} + (1-\lambda)\hat{P}_{\mathrm{tar}}$ with the mixture coefficient $\lambda \in [0, 1]$ and denote by 'mix' the induced MDP. Under this construction, the following gap bound holds, where the theorem is adapted from Xu et al. (2023).

**Theorem 1.** *Let $\eta_{\mathrm{tar}}(\pi)$ and $\eta_{\mathrm{mix}}(\pi)$ denote the expected returns of a policy $\pi$ in the target domain and in the proposed mixture domain, respectively. Then the performance gap between the two domains can be bounded from the perspectives of transition dynamics and value functions as follows.*

*Gap bound (transition dynamics).*

$$\eta_{\mathrm{mix}}(\pi) - \eta_{\mathrm{tar}}(\pi) \leq \frac{2\gamma r_{\max}}{(1-\gamma)^2}\Big(\lambda\,\mathbb{E}_{\rho_{\mathrm{mix}}^{\pi}}\big[D_{\mathrm{TV}}(P_{\mathrm{src}} \,\|\, P_{\mathrm{tar}})\big] + (1-\lambda)\,\mathbb{E}_{\rho_{\mathrm{mix}}^{\pi}}\big[D_{\mathrm{TV}}(\hat{P}_{\mathrm{tar}} \,\|\, P_{\mathrm{tar}})\big]\Big),$$

(3)

*where $D_{\mathrm{TV}}(P \,\|\, Q)$ denotes the total variation distance between $P$ and $Q$.*

*Gap bound (value discrepancy).*

$$\eta_{\mathrm{mix}}(\pi) - \eta_{\mathrm{tar}}(\pi) \leq \frac{\gamma}{(1-\gamma)}\Big(\lambda\,\mathbb{E}_{\rho_{\mathcal{M}_{\mathrm{mix}}}^{\pi}}\Big[\big|\mathbb{E}_{P_{\mathrm{src}}}\big[V_{\mathcal{M}_{\mathrm{tar}}}^{\pi}(s')\big] - \mathbb{E}_{P_{\mathrm{tar}}}\big[V_{\mathcal{M}_{\mathrm{tar}}}^{\pi}(s')\big]\big|\Big]$$
$$+ (1-\lambda)\,\mathbb{E}_{\rho_{\mathcal{M}_{\mathrm{mix}}}^{\pi}}\Big[\big|\mathbb{E}_{\hat{P}_{\mathrm{tar}}}\big[V_{\mathcal{M}_{\mathrm{tar}}}^{\pi}(s')\big] - \mathbb{E}_{P_{\mathrm{tar}}}\big[V_{\mathcal{M}_{\mathrm{tar}}}^{\pi}(s')\big]\big|\Big]\Big).$$

(4)

**Proof)** Proof of Theorem 1 is provided in Appendix H.

From Theorem 1, the performance gap bound in the mixture MDP can be reduced in two ways: (1) by reducing the discrepancy $D_{\mathrm{TV}}(P_{\mathrm{src}}, \|, P_{\mathrm{tar}})$ through selecting source samples that are close to the target distribution, as in existing distance based approaches, and (2) by reducing $D_{\mathrm{TV}}(\hat{P}_{\mathrm{tar}}, \|, P_{\mathrm{tar}})$ via learning a target transition estimator that closely approximates $P_{\mathrm{tar}}$ and then decreasing $\lambda$ to tighten the overall gap bound. In this work, we consider both directions. To address the second direction, we introduce the TCE method, which generates target-like transitions for mixture states constructed from both source and target data. Following the score-based generative modeling framework in equation 1, we first train a mixture-based state score network $q_{\theta}^{\mathrm{mix}}$ that expands state coverage over a controllable mixture of source and target states, and a target-transition score network $q_{\theta}^{\mathrm{tran}}$ that

generates transitions consistent with target-domain dynamics. The joint training objective for both score networks is defined as:

$$\underbrace{\mathbb{E}_{\tau,\, s_t \sim \mathcal{D}_{\mathrm{src}} \cup \mathcal{D}_{\mathrm{tar}}}\Big[\lambda(\tau)\big\| q_\theta^{\mathrm{mix}}(s_t^\tau, \tau \mid y(s_t)) + \tfrac{z}{\sigma(\tau)}\big\|_2^2\Big]}_{\textbf{mixture-based state score network loss}} + \underbrace{\mathbb{E}_{\tau,\, (s_t, s_{t+1}) \sim \mathcal{D}_{\mathrm{tar}}}\Big[\lambda(\tau)\big\| q_\theta^{\mathrm{tran}}(s_{t+1}^\tau, \tau \mid s_t) + \tfrac{z}{\sigma(\tau)}\big\|_2^2\Big]}_{\textbf{target-transition score network loss}},$$

(5)

where $z \sim \mathcal{N}(0, I)$, $s_t^\tau = \alpha(\tau)s_t + \sigma(\tau)z$, and $y(s_t)$ is a binary label indicating whether $s_t$ comes from source data $\mathcal{D}_{\mathrm{src}}$ ($y = 1$) or target data $\mathcal{D}_{\mathrm{tar}}$ ($y = 0$). Since the source and target domains share the same state space, the mixture-based state score network is trained on both datasets to broaden state coverage by conditioning on $y(s_t)$. While the label is deterministic during training, it is later treated as a continuous control parameter during sampling, allowing interpolation between the two domains and fine-grained adjustment of state-space coverage. The target-transition score network is trained solely on $\mathcal{D}_{\mathrm{tar}}$ to model next states that follow target-domain transition dynamics, enabling the construction of transitions for newly generated states that remain consistent with the target domain.

After training, we expand state–transition coverage using a two-stage sampling procedure based on the reverse SDE in equation 2. During sampling, we first draw a label parameter $\hat{y} \sim \mathrm{Unif}(0, y_{\max})$, where $y_{\max} \in (0, 1]$ is the label bound that specifies the maximum state-space coverage toward the source domain. Larger values of $\hat{y}$ result in broader state-space coverage by generating samples closer to the source dataset distribution, whereas smaller values bias generation toward the target dataset distribution, thereby reducing potential overfitting of the transition model trained on limited data.

**Stage 1 (State Sampling):** starting from Gaussian noise $s^K \sim \mathcal{N}(0, I)$, we integrate the reverse SDE backward from $k = K$ to 0 using the mixture-based score network conditioned on $\hat{y}$:

$$s^{k-1} = s^k + \Big[f(s^k, \tau^k) - g(\tau^k)^2 q_\theta^{\mathrm{mix}}(s^k, \tau^k \mid \hat{y})\Big]\Delta\tau^k + g(\tau^k)\sqrt{\Delta\tau^k}\,\xi^k, \quad \xi^k \sim \mathcal{N}(0, I), \quad (6)$$

where $f$ and $g$ denote the drift and diffusion coefficients in equation 2. After all steps, $s^0$ is taken as the generated state $\hat{s}_t$.

**Stage 2 (Transition Sampling):** conditioned on the generated state $\hat{s}_t$, we obtain its next state $\hat{s}_{t+1}$ by solving the same reverse SDE using the target-transition score network:

$$s^{k-1} = s^k + \Big[f(s^k, \tau^k) - g(\tau^k)^2 q_\theta^{\mathrm{tran}}(s^k, \tau^k \mid \hat{s}_t)\Big]\Delta\tau^k + g(\tau^k)\sqrt{\Delta\tau^k}\,\xi^k, \quad \xi^k \sim \mathcal{N}(0, I), \quad (7)$$

again integrating from $k = K$ to 0 to yield $\hat{s}_{t+1} = s^0$. The resulting pair $(\hat{s}_t, \hat{s}_{t+1})$ forms a synthetic transition that expands the support of target-domain transitions while remaining consistent with target dynamics, thus mitigating distributional mismatch. This allows us to build a large set of target-aligned transitions that improve policy learning under scarce target data. While this paper focuses on challenging cross-domain tasks with continuous state spaces and adopts conditional score networks for transition generation, this step can be seamlessly extended to image-based states by replacing the conditional model with an inpainting-based sampling mechanism (Lugmayr et al., 2022), enabling vision-based control tasks without modifying the overall TCE framework.

### 4.3 Transition Filtering

During sampling, we obtain $N$ state transitions $x := (\hat{s}_t, \hat{s}_{t+1})$, which may include unrealistic samples due to modeling errors. To prevent such outliers from degrading policy learning, we apply Z-score filtering (Chandola et al., 2009), a simple and effective method that retains $x$ only if

$$\left| \frac{x_d - \mu_d}{\sigma_d} \right| \leq z_{\mathrm{th}}, \quad \forall d, \quad (8)$$

where $\mu_d$ and $\sigma_d$ are empirical statistics of the generated dataset and $z_{\mathrm{th}}$ is the Z-score threshold. This step discards extreme samples while avoiding excessive bias toward the target distribution. After filtering, we construct full transitions from the remaining synthetic state transitions for performing offline RL. To this end, we train two auxiliary models using the target dataset $\mathcal{D}_{\mathrm{tar}}$: an inverse dynamics model $\mathrm{Inv}_\psi$ trained to predict the action given a state pair $(s_t, s_{t+1})$, and a reward model $R_\psi$ trained to estimate the reward directly from $(s_t, s_{t+1})$. Although rewards in many environments depend on actions, we follow Tian et al. (2024) and use only state pairs to predict rewards, since

Figure 2: The structure of TCE

**Algorithm 1** TCE Framework

1: **Input:** $\mathcal{D}_{\text{tar}}, \mathcal{D}_{\text{src}}, y_{\max}, z_{\text{th}}, \lambda$
2: **Train:** train $q_\theta^{\text{mix}}$ on $\mathcal{D}_{\text{src}} \cup \mathcal{D}_{\text{tar}}$ (domain label), and $q_\theta^{\text{tran}}$ on $\mathcal{D}_{\text{tar}}$
3: **Source selection:** set the source data $\mathcal{D}_{\text{src}}^{\text{NN},\lambda}$ based on equation 9
4: **Two-Stage Sampling:**
5: Stage 1: sample $\hat{y} \sim \text{Unif}[0, y_{\max}]$, generate $\hat{s}_t$ with $q_\theta^{\text{mix}}$; repeat to collect $N$ transitions
6: Stage 2: generate $\hat{s}_{t+1}$ with $q_\theta^{\text{tran}}$ conditioned on $\hat{s}_t$
7: **Label & Filter:** recover $\hat{a}_t$ and $\hat{r}_t$ using inverse and reward models, and remove outliers by Z-score with $z_{\text{th}}$
8: **Offline RL:** train $\pi$ using IQL on $\mathcal{D}_{\text{tar}} \cup \mathcal{D}_{\text{mix}}^\lambda$

actions inferred by the inverse model can be noisy. For each generated transition $(\hat{s}_t, \hat{s}_{t+1})$, we then recover $\hat{a}_t = \text{Inv}_\psi(\hat{s}_t, \hat{s}_{t+1})$ and $\hat{r}_t = R_\psi(\hat{s}_t, \hat{s}_{t+1})$, yielding the complete synthetic transition $(\hat{s}_t, \hat{a}_t, \hat{r}_t, \hat{s}_{t+1})$. Motivated by Theorem 1, we collect all such transitions and augment both the source and target datasets to construct the mixture dataset.

## 4.4 DATASET CONFIGURATION AND OFFLINE POLICY LEARNING

To make the training setup consistent with Theorem 1, we construct a mixture dataset in which source transitions and TCE generated transitions appear in the ratio $\lambda : (1 - \lambda)$. To further reduce the gap bound introduced by source data , we use only source transitions that are sufficiently close to the target data, and for simplicity we adopt an efficient nearest neighbor (NN) distance instead of more complex distance estimators. To do this, for each $(s, a, s') \in \mathcal{D}_{\text{src}}$, we define the NN distance $d_{\text{NN}}(s, a, s') := \min_{(s_{\text{tar}}, a_{\text{tar}}, s'_{\text{tar}}) \in \mathcal{D}_{\text{tar}}} \big\| [s, a, s'] - [s_{\text{tar}}, a_{\text{tar}}, s'_{\text{tar}}] \big\|$, and compute the $\lambda$-quantile threshold $d_{\lambda,\text{NN}}$, and select source transitions as

$$\mathcal{D}_{\text{src}}^{\text{NN},\lambda} := \{(s, a, s') \in \mathcal{D}_{\text{src}} : d_{\text{NN}}(s, a, s') \leq d_{\lambda,\text{NN}}\}. \tag{9}$$

In parallel, we generate $(1 - \lambda)|\mathcal{D}_{\text{src}}|$ transitions with TCE, denoted $\mathcal{D}_{\text{gen}}^{(1-\lambda)}$, and obtain the mixed training dataset $\mathcal{D}_{\text{mix}}^\lambda := \mathcal{D}_{\text{src}}^{\text{NN},\lambda} \cup \mathcal{D}_{\text{gen}}^{(1-\lambda)}$ with $|\mathcal{D}_{\text{mix}}^\lambda| = |\mathcal{D}_{\text{src}}|$.

For offline RL, we adopt Implicit Q-Learning (IQL) for fair comparison with prior work, although any standard offline RL method could be applied. To further stabilize training, we incorporate a KL regularization term following prior offline RL study (Lyu et al., 2025), which penalizes deviation from the target-domain behavior policy. The resulting policy objective is

$$\mathcal{L}_\pi = \mathcal{L}_\pi^{\text{IQL}} + \beta \, \mathbb{E}_{s \sim \mathcal{D}_{\text{tar}}}\big[D_{\text{KL}}(\hat{\pi}_{\text{b}}(\cdot|s) \,\|\, \pi(\cdot|s))\big], \tag{10}$$

where training samples are drawn from $\mathcal{D}_{\text{tar}} \cup \mathcal{D}_{\text{mix}}^\lambda$, $\mathcal{L}_\pi^{\text{IQL}}$ is the standard IQL policy loss, $D_{\text{KL}}$ denotes the Kullback–Leibler (KL) divergence, $\hat{\pi}_{\text{b}}$ is the empirical behavior policy of the target dataset, and $\beta > 0$ controls the strength of the regularization. For implementation, we consider two variants: TCE (OG), which uses only generated samples without any source data ($\lambda = 0$), and TCE (NN), which uses the mixture dataset with $0 < \lambda < 1$. This separation allows us to clearly determine whether incorporating source data is beneficial or not. The overall framework of TCE is illustrated in Fig. 2, and the full procedure is summarized in Algorithm 1. Further implementation details, including loss formulations and training configurations, are provided in Appendix B.

## 5 EXPERIMENTS

In this section, we evaluate the proposed TCE across diverse cross-domain setups and compare it with recent cross-domain offline RL algorithms. We also conduct ablation studies to analyze the contribution of each component and examine the sensitivity to key hyperparameters. All reported results are averaged over 5 random seeds with mean and standard deviation.

### 5.1 EXPERIMENTAL SETUP

We evaluate on cross-domain setups from MuJoCo continuous-control tasks (Todorov et al., 2012) as proposed by Lyu et al. (2025). The source and target domains share the same agent type (HalfCheetah,

| Src. | Tgt. | IQL* | DARA | BOSA | SRPO | IGDF | OTDF | TCE(OG) | TCE(NN) |
|---|---|---|---|---|---|---|---|---|---|
| half-m | m | 30.0±1.6 | 26.6±3.3 | 19.3±3.5 | 41.3±0.4 | 41.6±0.5 | 39.1±2.3 | **44.1±0.2** | 43.8±0.2 |
| half-m | m-e | 31.8±1.1 | 32.0±0.7 | 33.6±1.1 | 30.7±0.8 | 29.6±2.2 | 35.6±0.7 | **43.8±0.1** | 43.7±0.1 |
| half-m | e | 8.5±1.0 | 9.3±1.6 | 7.9±0.8 | 8.6±0.9 | 10.0±0.8 | 10.7±1.2 | 82.8±0.1 | **85.0±1.2** |
| half-m-r | m | 30.8±4.4 | 35.6±0.7 | 35.0±4.6 | 32.0±1.4 | 28.0±2.0 | 40.0±1.2 | **44.0±0.2** | 43.6±0.2 |
| half-m-r | m-e | 12.9±2.2 | 16.9±4.1 | 19.9±5.5 | 12.4±1.6 | 12.0±3.7 | 34.4±0.7 | **44.2±0.3** | 43.7±0.1 |
| half-m-r | e | 5.9±1.7 | 3.7±2.7 | 2.4±1.9 | 6.2±1.4 | 5.3±2.3 | 8.2±2.7 | **84.4±4** | 77.9±0.2 |
| half-m-e | m | 41.5±0.1 | 40.3±1.2 | 41.3±0.3 | 41.3±0.4 | 40.9±0.4 | 41.4±0.3 | **44.2±0.1** | 43.7±0.1 |
| half-m-e | m-e | 25.8±2.0 | 30.6±2.8 | 32.1±0.8 | 27.2±0.8 | 26.2±1.8 | 35.1±0.6 | 43.8±0.1 | **43.9±0.5** |
| half-m-e | e | 7.8±1.3 | 8.3±1.3 | 9.1±0.8 | 7.8±0.9 | 7.5±0.9 | 9.8±1.0 | **85.1±0.8** | 82.6±0.2 |
| hopp-m | m | 13.5±0.2 | 13.5±0.4 | 13.2±0.3 | 13.4±0.1 | 13.4±0.2 | 11.0±0.9 | **39.1±0.2** | 8.0±2.3 |
| hopp-m | m-e | 13.4±0.1 | 13.6±0.2 | 11.2±4.6 | 13.3±0.2 | 13.3±0.4 | 12.6±0.8 | **29.1±0.1** | 11.0±0.3 |
| hopp-m | e | 13.5±0.2 | 13.6±0.3 | 13.3±0.4 | 13.6±0.2 | 13.9±0.1 | 10.7±4.7 | **99.8±0.1** | 10.4±0.1 |
| hopp-m-r | m | 10.8±1.1 | 10.2±1.0 | 1.2±0.0 | 10.7±1.6 | 12.0±4.4 | 8.7±2.8 | **49.5±0.1** | 10.7±0.1 |
| hopp-m-r | m-e | 11.6±1.6 | 10.4±0.9 | 1.3±0.2 | 10.4±1.2 | 8.2±2.8 | 9.7±2.7 | **17.4±0.3** | 8.3±2.2 |
| hopp-m-r | e | 9.8±0.5 | 9.0±0.3 | 1.3±0.1 | 10.4±1.4 | 11.4±1.5 | 10.7±2.4 | **99.7±0.1** | 32.0±6.7 |
| hopp-m-e | m | 12.6±1.4 | 13.0±0.5 | 15.7±7.2 | 14.0±2.3 | 12.7±0.8 | 7.9±3.2 | **39.9±0.1** | 14.4±1.9 |
| hopp-m-e | m-e | **14.1±1.3** | 13.8±0.6 | 12.0±1.4 | 13.5±0.3 | 13.3±1.2 | 9.6±3.5 | 13.8±0.5 | 8.4±4.8 |
| hopp-m-e | e | 13.8±0.5 | 12.3±1.8 | 10.5±5.0 | 14.7±2.3 | 12.8±0.9 | 5.9±4.0 | **99.6±0.1** | 12.7±0.2 |
| walk-m | m | 23.0±4.7 | 23.3±3.3 | 6.2±2.9 | 24.7±1.7 | 27.5±9.5 | 50.5±5.8 | 44.2±0.2 | 38.1±2.1 |
| walk-m | m-e | 21.5±8.6 | 22.2±7.6 | 7.2±2.9 | 18.7±7.3 | 20.7±5.9 | **44.3±23.8** | 37.8±7.0 | 18.6±2.8 |
| walk-m | e | 20.3±2.8 | 17.3±3.4 | 15.8±8.7 | 21.1±7.2 | 15.8±4.5 | 55.3±8.3 | 80.1±5.5 | **86.3±7.3** |
| walk-m-r | m | 11.3±3.0 | 10.9±4.6 | 5.4±4.0 | 10.4±4.8 | 13.4±7.2 | 37.4±5.1 | **43.5±3.7** | 37.4±0.1 |
| walk-m-r | m-e | 7.0±1.5 | 4.5±1.1 | 4.0±2.2 | 4.9±1.7 | 6.9±2.2 | 33.8±6.9 | **34.6±9.3** | 23.8±3.7 |
| walk-m-r | e | 6.3±0.9 | 4.5±1.1 | 3.8±3.4 | 5.5±0.9 | 5.5±2.2 | 41.5±6.8 | **74.8±0.4** | 50.9±17.5 |
| walk-m-e | m | 24.1±7.4 | 31.7±6.6 | 18.7±6.5 | 29.9±4.7 | 27.5±2.3 | **49.9±4.6** | 41.3±1.2 | 42.0±0.1 |
| walk-m-e | m-e | 27.0±5.5 | 23.3±5.5 | 11.1±0.9 | 22.9±3.8 | 25.3±6.4 | **40.5±11.0** | 32.9±5.1 | 27.3±4.1 |
| walk-m-e | e | 22.4±3.3 | 25.2±5.7 | 9.9±3.9 | 18.7±5.7 | 24.7±2.4 | 45.7±6.9 | 75.9±7.5 | **84.4±2.9** |
| ant-m | m | 38.7±3.8 | 41.3±1.8 | 18.2±1.9 | 40.6±2.1 | 40.9±1.7 | 39.4±1.7 | **41.8±0.7** | 41.3±0.3 |
| ant-m | m-e | 47.0±5.1 | 43.3±2.0 | 45.3±7.0 | 47.2±4.3 | 44.4±1.7 | 58.3±8.9 | **73.8±1.9** | 71.1±2.5 |
| ant-m | e | 36.2±3.5 | 48.5±4.2 | 72.2±10.5 | 42.2±9.9 | 41.4±4.2 | 85.4±4.4 | 93.6±1.3 | **95.6±1.1** |
| ant-m-r | m | 38.2±2.9 | 38.9±2.7 | 20.2±3.7 | 38.3±1.9 | 39.7±1.2 | **41.2±0.9** | 41.2±0.6 | 40.7±0.1 |
| ant-m-r | m-e | 38.1±3.5 | 33.4±5.5 | 15.2±1.6 | 35.0±5.7 | 37.3±2.4 | 50.8±4.5 | **74.3±1.6** | 72.7±4.1 |
| ant-m-r | e | 24.1±1.9 | 24.5±2.6 | 16.0±1.7 | 22.7±3.0 | 23.6±1.4 | 67.2±7.5 | **91.9±0.3** | 81.0±2.1 |
| ant-m-e | m | 32.9±5.1 | 40.2±1.5 | 28.1±5.6 | 35.9±2.5 | 36.1±4.4 | 39.9±2.9 | **41.5±0.1** | 37.6±0.1 |
| ant-m-e | m-e | 35.7±3.9 | 36.5±8.7 | 14.8±15.9 | 24.5±15.7 | 30.7±10.8 | 65.7±4.5 | 72.1±5.5 | **72.5±4.3** |
| ant-m-e | e | 36.1±8.5 | 34.6±5.8 | 53.9±5.0 | 38.4±9.4 | 35.2±6.6 | 86.4±2.2 | 93.9±1.3 | **94.3±1.6** |
| **Total Score** | | 798.0 | 816.8 | 646.3 | 803.1 | 808.7 | 1274.3 | **2093.5** | 1639.4 |

Table 1: Performance comparison on 36 morphology-shift tasks. Abbrev.: half=HalfCheetah, hopp=Hopper, walk=Walker2d, ant=Ant; m=medium, m-r=medium-replay, e=expert, m-e=medium-expert. Src./Tgt. denote source/target domain, respectively. Results are reported as mean ± standard deviation over 5 seeds, with the best result in each row shown in **bold**, and the second best result is underlined.

Hopper, Walker2d, Ant) but differ in morphology, kinematics, or gravity parameters, creating large domain gaps. In the offline setting, each domain uses pre-collected datasets of varying quality from the D4RL benchmark (Fu et al., 2020), widely used for offline RL. This setup uses different dataset qualities for the source and target domains, creating a more challenging setup. For the target domain, we consider 3 datasets: expert, obtained from a fully converged expert policy, medium, obtained from a partially trained policy, and medium-expert, a mixture of medium and expert data. For the source domain, we use 3 datasets: medium, medium-replay, obtained from the replay buffer during medium-policy training; and medium-expert. This setup yields 36 cross-domain tasks for each morphology, kinematics, and gravity shifts setup. The source dataset contains roughly 1M–2M transitions, whereas the target dataset is restricted to 5k transitions, reflecting the difficulty of collecting target-domain data. All results are reported as normalized returns, where 0 corresponds to a random policy and 100 to an expert policy. Further details of the environmental setup are provided in Appendix C.

## 5.2 PERFORMANCE COMPARISON

For performance comparison, we evaluate TCE against a comprehensive set of cross-domain offline RL baselines: **IQL\*** (Kostrikov et al., 2022), which trains IQL on the union of source and target data; **DARA** (Liu et al., 2022), which employs domain-adversarial classifiers to mitigate dynamics mismatch; **BOSA** (Liu et al., 2024), which constrains the policy to the support of the dataset; **SRPO** (Xue et al., 2023), which regularizes policy learning by matching stationary distributions; **IGDF** (Wen et al., 2024), which filters source transitions using contrastive representation learning; and **OTDF** (Lyu et al., 2025), which performs source filtering based on optimal transport distances. For the baseline methods, we report results directly from (Lyu et al., 2025), which implemented each algorithm with hyperparameter tuning. For TCE, we consider TCE(OG) and TCE(NN). For TCE(NN),

| Src. | Tgt. | IQL* | DARA | BOSA | SRPO | IGDF | OTDF | TCE(OG) | TCE(NN) |
|------|------|------|------|------|------|------|------|---------|---------|
| half-m | m | 12.3±1.2 | 10.6±1.2 | 8.3±1.2 | 16.8±4.2 | 23.6±5.7 | 40.2±0.0 | **41.9±0.9** | 41.4±0.1 |
| half-m | m-e | 10.8±1.9 | 12.9±2.8 | 8.7±1.3 | 10.3±2.7 | 9.8±2.4 | 10.1±4.0 | 39.7±1.1 | **40.5±0.5** |
| half-m | e | 12.6±1.7 | 12.1±1.0 | 10.8±1.7 | 12.2±0.9 | **12.8±0.7** | 8.7±2.0 | 11.9±4.6 | 7.5±1.1 |
| half-m-r | m | 10.0±5.4 | 11.5±4.9 | 7.5±3.1 | 10.2±3.7 | 11.6±4.6 | 37.8±2.1 | **41.8±0.5** | 40.2±0.9 |
| half-m-r | m-e | 6.5±3.1 | 9.2±4.7 | 6.6±1.7 | 9.5±1.8 | 8.6±2.3 | 9.7±2.0 | **40.8±1.4** | 33.6±6.4 |
| half-m-r | e | 13.6±1.4 | 14.8±2.0 | 10.4±4.9 | 14.8±2.2 | 13.9±2.2 | 7.2±1.4 | **15.2±6.4** | 2.9±0.1 |
| half-m-e | m | 21.8±6.5 | 25.9±7.4 | 30.0±4.3 | 17.2±3.3 | 21.9±6.5 | 30.7±9.6 | **42.0±0.2** | 41.1±0.5 |
| half-m-e | m-e | 7.6±1.4 | 9.5±4.2 | 6.8±2.9 | 9.6±2.4 | 8.9±3.3 | 10.9±4.2 | **41.2±0.6** | 35.8±1.8 |
| half-m-e | e | 9.1±2.4 | 10.4±1.3 | 4.9±3.2 | **11.2±1.0** | 10.7±1.4 | 3.2±0.6 | 9.5±7.4 | 7.5±1.6 |
| hopp-m | m | 58.7±8.4 | 43.9±15.2 | 12.3±6.6 | 65.4±1.5 | 65.3±1.4 | 65.6±1.9 | **66.8±0.5** | 66.3±0.2 |
| hopp-m | m-e | 68.5±12.4 | 55.4±16.9 | 15.6±10.8 | 43.9±30.8 | 51.1±18.5 | 55.4±25.1 | **72.1±4.1** | 67.3±2.9 |
| hopp-m | e | 79.9±35.5 | 83.7±19.6 | 14.8±5.5 | 53.1±39.8 | 87.4±25.4 | 35.0±19.4 | **91.5±6.3** | 78.2±17.6 |
| hopp-m-r | m | 36.0±0.1 | 39.4±7.2 | 3.2±2.6 | 36.1±0.2 | 35.9±2.4 | 35.5±12.2 | 65.1±0.9 | **66.2±0.2** |
| hopp-m-r | m-e | 36.1±0.1 | 34.1±3.6 | 4.4±2.8 | 36.0±0.1 | 36.1±0.1 | 47.5±14.6 | **72.0±3.7** | 63.9±14.3 |
| hopp-m-r | e | 36.0±0.1 | 36.1±0.2 | 3.7±2.5 | 36.1±0.1 | 36.1±0.3 | 49.9±30.5 | **96.8±2.4** | 85.1±2.5 |
| hopp-m-e | m | 66.0±0.5 | 61.1±4.0 | 35.0±20.1 | 64.6±2.6 | 65.2±1.5 | 65.3±2.4 | **66.6±0.6** | 66.2±0.2 |
| hopp-m-e | m-e | 45.1±15.7 | 61.9±16.9 | 13.9±4.9 | 54.7±17.0 | 62.9±15.6 | 38.6±15.9 | **76.0±2.0** | 72.7±3.1 |
| hopp-m-e | e | 44.9±19.8 | 84.2±21.1 | 12.0±4.3 | 57.6±40.6 | 52.8±39.7 | 29.9±11.3 | 89.2±8.4 | **89.7±4.2** |
| walk-m | m | 34.3±9.8 | 35.2±22.5 | 14.3±11.2 | 39.0±6.7 | 41.9±11.2 | 49.6±18.0 | **60.4±1.9** | 54.1±2.1 |
| walk-m | m-e | 30.2±12.5 | **51.9±11.5** | 13.6±7.7 | 38.6±6.5 | 42.3±19.3 | 43.5±16.4 | 46.2±12.1 | 19.8±1.3 |
| walk-m | e | 56.4±18.2 | 40.7±14.4 | 15.3±2.5 | 57.3±12.2 | **60.4±17.5** | 46.7±13.6 | 59.3±4.2 | 33.4±1.5 |
| walk-m-r | m | 11.5±7.1 | 12.5±4.3 | 1.9±2.1 | 14.3±3.1 | 22.2±5.2 | 49.7±9.7 | **50.2±3.7** | 45.1±4.5 |
| walk-m-r | m-e | 9.7±3.8 | 11.2±5.0 | 4.6±3.0 | 4.2±5.1 | 7.6±4.9 | **55.9±17.1** | 37.1±11.8 | 21.3±5.5 |
| walk-m-r | e | 7.7±4.8 | 7.4±2.4 | 3.6±1.5 | 13.2±8.5 | 7.5±2.1 | 51.9±7.9 | **53.0±7.9** | 23.1±2.9 |
| walk-m-e | m | 41.8±8.8 | 38.1±14.4 | 21.4±8.3 | 36.9±4.3 | 41.2±13.0 | 44.6±6.0 | **55.2±2.5** | 54.9±3.2 |
| walk-m-e | m-e | 22.2±8.7 | 23.6±8.1 | 15.9±4.1 | 23.2±7.9 | 28.1±4.0 | 16.5±7.2 | **31.2±4.8** | 24.7±3.8 |
| walk-m-e | e | 26.3±10.4 | 36.0±9.2 | 18.5±3.6 | 40.9±9.6 | 46.2±19.4 | 42.4±9.1 | **47.1±18.1** | 25.2±6.1 |
| ant-m | m | 50.0±5.6 | 42.3±7.6 | 20.9±2.6 | 50.5±6.7 | 54.5±1.3 | **55.4±0.0** | 53.2±1.9 | 47.5±1.9 |
| ant-m | m-e | 57.8±7.2 | 54.1±3.8 | 31.7±7.0 | 54.9±1.3 | 54.5±4.6 | 60.7±3.6 | 61.4±2.0 | **64.2±9.5** |
| ant-m | e | 59.6±18.5 | 54.2±11.3 | 45.4±8.6 | 45.5±9.3 | 49.4±14.6 | 90.4±4.8 | 92.7±2.8 | **93.8±3.4** |
| ant-m-r | m | 43.7±4.6 | 42.0±5.4 | 19.0±1.8 | 45.3±5.1 | 41.4±5.0 | 52.8±4.4 | **54.6±1.4** | 51.0±3.2 |
| ant-m-r | m-e | 36.5±5.9 | 36.0±6.7 | 19.1±1.6 | 36.2±6.6 | 37.2±4.7 | 54.2±5.2 | 61.6±2.4 | **61.7±5.4** |
| ant-m-r | e | 24.4±4.8 | 22.1±0.4 | 19.5±0.8 | 27.1±3.7 | 24.3±2.8 | 74.7±10.5 | 92.0±2.4 | **94.2±0.2** |
| ant-m-e | m | 49.5±4.1 | 44.7±4.3 | 19.0±8.0 | 41.3±8.1 | 41.8±8.8 | 50.2±4.3 | **55.6±1.4** | 55.1±3.7 |
| ant-m-e | m-e | 37.2±2.0 | 33.3±7.0 | 6.4±2.5 | 38.2±8.0 | 41.5±4.9 | 48.8±2.7 | 59.1±3.4 | **62.1±0.2** |
| ant-m-e | e | 18.7±8.1 | 17.8±23.6 | 14.5±9.0 | 35.2±15.5 | 14.4±22.9 | 78.4±12.2 | **94.2±3.2** | 90.3±1.3 |
| **Total Score** | | 1193.0 | 1219.8 | 513.5 | 1195.7 | 1271.0 | 1547.6 | **2044.2** | 1828.1 |

Table 2: Performance comparison in cross-domain offline RL under kinematic shifts.

we use the best $\lambda$ over $0.1 \leq \lambda \leq 0.9$. All other hyperparameters are chosen via hyperparameter search, and TCEs are trained for the same number of steps as the baselines. More experimental details are provided in Appendix C, and we also present the results for morphology and kinematic shifts in the main text and provide the results for gravity shifts in Appendix D.1.

**Morphology Shifts.** Table 1 summarizes the results under morphology shifts. TCE methods achieve the highest average performance on 31 of 36 tasks, significantly outperforming all baselines. The performance gain is most pronounced when the target dataset is of high quality, such as expert, where most baselines fail to learn effectively due to the narrow state distribution and large domain gap. By combining controllable state coverage expansion with target-aligned transition generation, our method mitigates distributional shift and enables effective policy learning even in these challenging settings. In addition, morphology shift typically induces a large domain gap between source and target. In such cases, TCE(OG) is usually clearly superior, and TCE(NN) attains its best results when the mixing weight $\lambda$ is as small as 0.1. As a result, the performance gap between TCE(OG) and TCE(NN) is generally small, and in some settings even a slight use of source data hurts performance, indicating that under large domain gaps source-only methods perform much worse than TCE(OG) and that using only generated samples can be more effective than mixing in source data.

**Kinematic Shifts & Gravity Shifts.** Table 2 reports the results for kinematic shifts. Although these shifts involve milder changes in dynamics compared to morphology shifts, TCE methods still achieve the highest average return across nearly all tasks. Kinematic shift has a smaller domain gap than morphology shift. In some environments TCE(NN) slightly outperforms TCE(OG), but in most cases it still lags behind, consistent with the morphology results. In contrast, the gravity shift results in Table D.1 of Appendix D.1 reflect an even smaller domain gap. In this regime, TCE(OG) degrades substantially, whereas TCE(NN), which mixes source data, clearly surpasses both TCE(OG) and other baselines. This shows that when the domain gap is small and target data are limited, exploiting source data is beneficial, and TCE methods still remain stronger than the baselines.

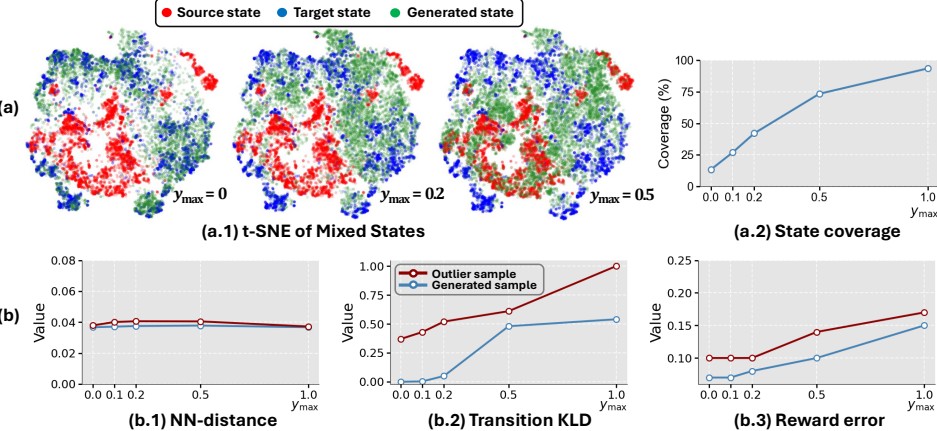

**(a.1) t-SNE of Mixed States**

**(a.2) State coverage**

**(b.1) NN-distance**

**(b.2) Transition KLD**

**(b.3) Reward error**

Figure 3: Coverage and sample reliability with respect to $y_{\max}$ in Ant morphology shifts. (a.1) t-SNE visualization of state datasets and (a.2) the corresponding coverage curve. (b.1) NN-distance between generated states $\mathcal{D}_{\mathrm{gen}}^{1-\lambda}$ and true states in $\mathcal{D}_{\mathrm{tar}} \cup \mathcal{D}_{\mathrm{src}}$. (b.2) Transition KL divergence(normalized) and (b.3) reward error between models trained on limited and sufficient target data.

## 5.3 COVERAGE AND SAMPLE RELIABILITY ANALYSIS

To better understand how the proposed method enhances coverage, improves sample reliability, and leverages Z-score filtering, Fig. 3(a) shows a t-SNE visualization of the source, target, and TCE-generated data together with the corresponding coverage curve as a function of $y_{\max}$, while Fig. 3(b) compares the errors of generated and Z-score-filtered outlier samples with respect to states, transitions, and rewards as $y_{\max}$ varies. In terms of coverage, as $y_{\max}$ increases, the generated states smoothly interpolate between the two domains: when $y_{\max} = 0$, the samples closely match the target distribution, whereas larger values yield samples resembling the source distribution, thereby broadening coverage as intended. In contrast, for sample reliability, increasing $y_{\max}$ can hurt generalization given the limited target data. In Fig. 3(b), the state error is the discrepancy between generated states and true states from the source and target datasets, while the transition KL divergence and reward error compare models trained on limited versus abundant target data. The results show that increasing $y_{\max}$ does not substantially increase the state error, so the generated states themselves remain reliable; however, once $y_{\max} > 0.2$, both transition and reward errors rise sharply, which degrades generalization. We therefore regard generated transitions as trustworthy up to $y_{\max} = 0.2$. The figure also reports the errors of samples rejected by Z-score filtering, which are much larger across $y_{\max}$, confirming that the proposed filtering effectively removes low-quality samples. Appendix F provides additional reliability analyses in other environments.

## 5.4 ABLATION STUDY

In this section, we analyze the contribution of each component through component-wise evaluation and study the effect of key hyperparameters, namely the label bound $y_{\max}$ and the Z-score threshold $z_{\mathrm{th}}$. Additional analysis of computational complexity and further ablations on the denoising step $K$ are provided in Appendices E and G.

**Component Evaluation** We ablate five configurations in increasing methodological completeness.

Table 3: Component evaluation on morphology shifts: Average normalized return over 36 tasks

| Setting | Average Scores |
|---|---|
| TCE(OG) | **58.2±23.8** |
| TCE(NN) | **45.5±27.5** |
| TCE(OT) | **48.2±26.1** |
| TCE(OG) w/o Policy Reg. | **55.8±26.8** |
| TCE(OG) w/o Filtering | 56.7±27.5 |
| Simple Aug. | 45.9±22.6 |
| Target+Source(whole) | 21.3±12.1 |
| Target Only | 41.4±21.2 |

To evaluate the contribution of each component, we consider six configurations with increasing methodological completeness. TCE(OT) adopts the optimal-transport distance of Lyu et al. (2025) for source selection; TCE(OG) w/o Filtering removes the Z-score filtering step; TCE(OG) w/o Policy Reg. omits the policy regularization term; Simple Augmentation augments state transitions using only 0.5M target-domain transitions; Target+Source(whole) trains IQL on the naive union of source and target data (equivalent to IQL* in Table 1); and Target Only trains IQLusing only 5k target-domain samples. . Table 3 reports the average normalized return over 36 morphology-shift tasks. TCE(OG) achieves

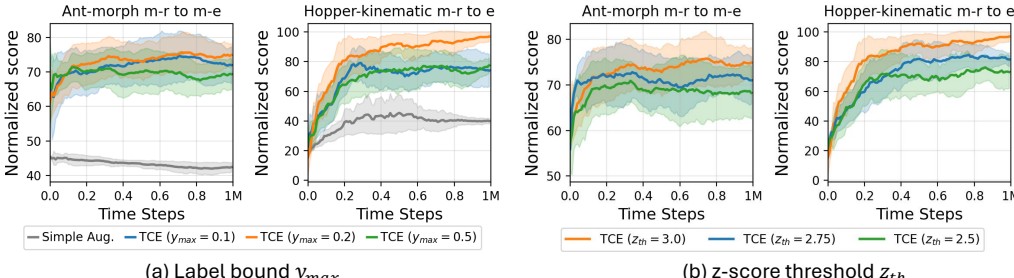

(a) Label bound $y_{max}$        (b) z-score threshold $z_{th}$

Figure 4: Hyperparameter analysis on (a) Ant-morphology and (b) Hopper-kinematics tasks: (a) effect of label bound $y_{\max}$, (b) effect of Z-score threshold $z_{\mathrm{th}}$

the highest normalized return, showing that its components are crucial and work synergistically. In particular, TCE(OG) significantly outperforms both Simple Augmentation (which uses only target data) and Target+Source, demonstrating that our approach effectively expands state coverage while minimizing distributional mismatch, leading to improved policy performance. In addition, TCE(OT) achieves slightly higher scores than TCE(NN) but still falls short of TCE(OG), indicating that TCE-based sample generation is considerably more important than source selection alone. The results of TCE(OG) w/o Policy Reg. further show that removing policy regularization degrades performance, although the proposed method remains effective.

**Label Bound** $y_{\mathrm{max}}$**:** The hyperparameter $y_{\mathrm{max}}$ controls how strongly the mixture-based score network $q_\theta^{\mathrm{mix}}$ shifts state generation toward the source dataset, thereby determining the overall state-space coverage. As shown in Fig. 4(a) and consistent with the sampling analysis in Fig. 3, coverage increases monotonically with $y_{\mathrm{max}}$, but excessively large values such as $y_{\mathrm{max}} = 0.5$ produce states that deviate too far from the target distribution. This leads to larger transition-model errors, as observed in the KL-divergence analysis, and ultimately degrades policy performance. We find that $y_{\mathrm{max}} = 0.2$ achieves the best trade-off, expanding coverage enough to improve policy learning while maintaining transition fidelity. Conversely, very small values such as $y_{\mathrm{max}} = 0.1$ or the Simple Augmentation setting without coverage expansion yield lower returns, underscoring the importance of controlled coverage expansion for the effectiveness of TCE.

**Z-score Threshold** $z_{\mathrm{th}}$**:** The Z-score filtering parameter $z_{\mathrm{th}}$ determines which samples are retained by discarding those whose Z-score exceeds $z_{\mathrm{th}}$, i.e., samples more than $z_{\mathrm{th}}$ standard deviations away from the mean. Fig. 4(b) shows that $z_{\mathrm{th}} = 3$ consistently achieves the best performance. When $z_{\mathrm{th}}$ is too low, for example 2.5 or 2.75, many samples far from the mean are removed, reducing diversity and diminishing the benefit of state coverage expansion. In contrast, values larger than 3 behave almost like no filtering, allowing unrealistic outliers to remain. Setting $z_{\mathrm{th}} = 3$ provides a balanced trade-off, filtering implausible samples while preserving enough diversity to improve policy learning.

## 6 LIMITATIONS

While TCE consistently outperforms strong baselines, it has two main limitations. First, because TCE trains two score networks and performs two-stage sampling, it introduces additional computational overhead compared to methods that simply reuse or filter source data. As analyzed in Appendix E, this overhead amounts to only a few extra hours in our setup, which is acceptable in the offline RL setting where the primary goal is to learn a high-quality policy without distributional shift rather than minimize training time. Second, TCE involves a few hyperparameters, such as the label bound and Z-score threshold, which control coverage and filtering. In practice, we find that performance is not highly sensitive to these parameters: moderate values consistently balance coverage and accuracy, and coverage expansion almost always improves performance, making TCE relatively easy to tune.

## 7 CONCLUSION

We presented TCE, a two-stage score-based framework that first expands target state coverage through mixture-conditioned sampling and then generates transitions aligned with target dynamics using a target-only score model, followed by conservative filtering. This design directly addresses distributional mismatch in cross-domain offline RL and enables effective policy learning with limited target data and no online interaction. Experiments on diverse MuJoCo domain shifts show that TCE consistently improves performance over prior methods, and ablations confirm the importance of both coverage expansion and target-aligned transition generation. These findings highlight TCE as a simple and practical solution for cross-domain offline RL.

ETHICS STATEMENT

This work proposes TCE for cross-domain offline reinforcement learning and focuses on improving methodology rather than real-world deployment. We do not identify any negative ethical concerns or potential negative social impacts associated with this research. The study does not involve human participants or personally identifiable data, and thus poses no safety or privacy risks.

REPRODUCIBILITY STATEMENT

We made significant efforts to ensure the reproducibility of our results. All datasets used in our experiments are publicly available, and a detailed description is provided in Appendix C.2. Our method is described in detail in Section 4.2 and Appendix B.2, with hyperparameter settings reported in Appendix C.3. All experiments are run with multiple random seeds, and we report mean and standard deviation for all results.

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

# A    THE USE OF LARGE LANGUAGE MODELS

In this work, we utilize LLMs solely to refine the manuscript, focusing on typographical corrections and improving readability. We did not use LLMs for research-related tasks such as idea formulation, methodological design, or result interpretation. All scientific contributions, experiments, and analyses were conducted entirely by the authors.

# B    DETAILED IMPLEMENTATION AND ALGORITHM OF TCE

This section summarizes core components of our proposed TCE method: Subsection B.1 defines the score-based training loss and reverse-time sampling with Langevin corrector. Subsection B.2 details the joint training of dual score networks, two-stage sampling, Z-score filtering, and synthetic transition labeling using inverse dynamics and reward models. Subsection B.3 provides network architecture and configuration details supporting the overall implementation.

## B.1    DETAILS OF SCORE-BASED GENERATIVE MODEL WITH SDEs

Our generative framework is built upon score-based models formulated through Stochastic Differential Equations (SDEs). We first define the noise schedule that governs the forward diffusion process. A clean data sample $x^0$ is perturbed over a continuous time variable $\tau \in [0,1]$ into a noisy sample $x^\tau = x^0 + \sigma(\tau)z$, where $z \sim \mathcal{N}(0, I)$ and the noise scale $\sigma(\tau)$ is given by:

$$\sigma(\tau) = \sqrt{1 - \exp\left(-\mathcal{B}(\tau)\right)}, \qquad \text{where} \quad \mathcal{B}(\tau) = \alpha_{\min}\tau + \frac{1}{2}(\alpha_{\max} - \alpha_{\min})\tau^2. \qquad \text{(B.1)}$$

Here, $\alpha_{\min}$ and $\alpha_{\max}$ control the minimum and maximum rates of noise injection, respectively. This quadratic schedule allows for a smooth, gradual increase in noise, which is beneficial for model training and sample quality. In all our experiments, we fix these values at $\alpha_{\min} = 0.1$ and $\alpha_{\max} = 20$.

**Training Objective of Generative Model:**    The core training objective is to learn a score network, $q_\theta(x^\tau, \tau \mid c)$, that estimates the gradient of the log-density of the noisy data, $\nabla_{x^\tau} \log p(x^\tau \mid c)$. The network is optimized via the denoising score matching loss:

$$\mathcal{L}_{\text{score}}(\theta) = E_{\tau,(x^0,c)}\left[\lambda(\tau)\left\|q_\theta(x^\tau, \tau \mid c) + \frac{z}{\sigma(\tau)}\right\|_2^2\right], \qquad z \sim \mathcal{N}(0, I) \qquad \text{(B.2)}$$

where the weighting function is chosen as $\lambda(\tau) = \sigma(\tau)^2$.

**Data Sampling:**    At inference time, samples are generated by solving the corresponding reverse-time SDE. We discretize the continuous time $\tau$ into a sequence of steps $1 = \tau^K > \cdots > \tau^0 = 0$, where $K$ is the denoising steps which fixed to $K = 500$ for all environments in this work. In its general discretized form, each reverse step is:

$$x^{k-1} = x^k + \left[f(x^k, \tau^k) - g(\tau^k)^2 q_\theta(x^k, \tau^k \mid c)\right]\Delta\tau^k + g(\tau^k)\sqrt{\Delta\tau^k}\,\xi^k, \quad \xi^k \sim \mathcal{N}(0, I). \text{ (B.3)}$$

In our implementation, we adopt the Variance Exploding (VE) SDE formulation, where the drift coefficient satisfies $f(x, \tau) = 0.$ , making the forward process purely noise-driven and thus simplifying sampling. The diffusion coefficient is defined as $g(\tau) = \sqrt{d(\sigma^2)/d\tau}$. For generation, we use Predictor–Corrector (PC) sampling, where the predictor integrates the reverse SDE and the corrector performs Langevin refinement, improving robustness to step size and noise levels and yielding higher-quality samples.

$$\text{(predictor step:)}\ \ x^{k-1} = x^k - g(\tau^k)^2 q_\theta(x^k, \tau^k \mid c)\Delta\tau^k + g(\tau^k)\sqrt{\Delta\tau^k}\,\xi^k \qquad \text{(B.4)}$$

After each predictor step, we apply a Langevin corrector step to refine sample quality:

$$\text{(corrector step:)}\ \ x^{k-1} \leftarrow x^{k-1} + \eta^k q_\theta(x^{k-1}, \tau^{k-1} \mid c) + \sqrt{2\eta^k}\zeta^k, \quad \zeta^k \sim \mathcal{N}(0, I) \qquad \text{(B.5)}$$

where $\eta^k$ is an adaptive step size that depends on the signal-to-noise ratio at step $k$. In all our experiments, the corrector step is applied once after each predictor step. Once PC sampling is completed, we take the sample obtained at the final time $\tau^0 = 0$, $x^0$ as the final sample.

## B.2 DETAILED IMPLEMENTATION OF TCE

We implement the proposed Two-stage Coverage Expansion (TCE) by jointly training two conditional score networks over noisy states and transitions.

**Training Objective of Score Networks in TCE:** The training objective for the two conditional score model $q_\theta^{\text{mix}}$ and $q_\theta^{\text{tran}}$ is defined as the sum of two denoising score matching losses based on equation B.2:

$$\mathcal{L}(\theta) = \mathbb{E}_{\tau, s_t \sim \mathcal{D}_{\text{src}} \cup \mathcal{D}_{\text{tar}}} \left[ \lambda(\tau) \left\| q_\theta^{\text{mix}}(s_t^\tau, \tau \mid y(s_t)) + \frac{z}{\sigma(\tau)} \right\|_2^2 \right]$$

$$+ \mathbb{E}_{\tau, (s_t, s_{t+1}) \sim \mathcal{D}_{\text{tar}}} \left[ \lambda(\tau) \left\| q_\theta^{\text{tran}}(s_{t+1}^\tau, \tau \mid s_t) + \frac{z}{\sigma(\tau)} \right\|_2^2 \right], \tag{B.6}$$

where $z \sim \mathcal{N}(0, I)$, and $s_t^\tau = s_t + \sigma(\tau)z$ denotes the noisy state at noise level $\tau$. Here, $y(s_t) \in \{0, 1\}$ is a binary domain label indicating whether $s_t$ is from the source or target domain.

We train an inverse dynamics model $\text{Inv}_\psi$ and a reward model $R_\psi$ on the target domain dataset $\mathcal{D}_{\text{tar}}$ using the combined mean squared error loss:

$$\mathcal{L}_{\text{inv}}(\psi) = \underbrace{\mathbb{E}_{(s_t, a_t, s_{t+1}) \sim \mathcal{D}_{\text{tar}}} \|\text{Inv}_\psi(s_t, s_{t+1}) - a_t\|_2^2}_{\text{Inverse dynamics loss}} + \underbrace{\mathbb{E}_{(s_t, r_t, s_{t+1}) \sim \mathcal{D}_{\text{tar}}} \|R_\psi(s_t, s_{t+1}) - r_t\|_2^2}_{\text{Reward prediction loss}}. \tag{B.7}$$

**Data Generation through TCE:** The two-stage coverage expansion utilize the two score model at each stage; $q_\theta^{\text{mix}}$ for sampling high-coverage states between source and target domain, and $q_\theta^{\text{tran}}$ for sampling their corresponding next states on target domain.

*Stage 1 (State Sampling):* Starting from $s_t^K \sim \mathcal{N}(0, I)$, we iteratively apply PC sampling according to Appendix B.1

(predictor step:) $s_t^{k-1} = s_t^k + \left[-g(\tau^k)^2 q_\theta^{\text{mix}}(s_t^k, \tau^k \mid \hat{y})\right] \Delta\tau^k + g(\tau^k)\sqrt{\Delta\tau^k}\xi^k$ (B.8)

(corrector step:) $s_t^{k-1} \leftarrow s_t^{k-1} + \eta^k q_\theta^{\text{mix}}(s_t^{k-1}, \tau^{k-1} \mid \hat{y}) + \sqrt{2\eta^k}\zeta^k$ (B.9)

Here, $\hat{y} \sim \text{Unif}(0, y_{\max})$ controls the mixture ratio. We take the sample obtained at the final time $\tau^0 = 0$, $s_t^0 = \hat{s}_t$ as the final generated state samples.

*Stage 2 (Transition Sampling):* Starting from $s_{t+1}^K \sim \mathcal{N}(0, I)$ and conditioned on the generated state $\hat{s}_t$, we iteratively apply PC sampling according to Appendix B.1

(predictor step:) $s_{t+1}^{k-1} = s_{t+1}^k + \left[-g(\tau^k)^2 q_\theta^{\text{tran}}(s_{t+1}^k, \tau^k \mid \hat{s}_t)\right] \Delta\tau^k + g(\tau^k)\sqrt{\Delta\tau^k}\xi^k$ (B.10)

(corrector step:) $s_{t+1}^{k-1} \leftarrow s_{t+1}^{k-1} + \eta^k q_\theta^{\text{tran}}(s_{t+1}^{k-1}, \tau^{k-1} \mid \hat{s}_t) + \sqrt{2\eta^k}\zeta^k$ (B.11)

We take the sample obtained at the final time $\tau^0 = 0$, $s_{t+1}^0 = \hat{s}_{t+1}$ as the final generated next state samples. Finally we get coverage expanded state transition samples $(\hat{s}_t, \hat{s}_{t+1})$.

After the two-stage coverage expansion data sampling, we apply Z-score filtering to remove extreme $(\hat{s}_t, \hat{s}_{t+1})$ outliers according to Section 4.3, by discarding the indices of them where the statistics of each dimensions exceed $z_{\text{th}}$. To label the filtered transitions $(\hat{s}_t, \hat{s}_{t+1})$ with corresponding actions and rewards, we then generate labels $(\hat{a}_t, \hat{r}_t) = (\text{Inv}_\psi(\hat{s}_t, \hat{s}_{t+1}), R_\psi(\hat{s}_t, \hat{s}_{t+1}))$ for each synthetic state pair using the $\text{Inv}_\psi$ and $R_\psi$ models. Finally we get the fully labeled dataset $\mathcal{D}_{\text{gen}}$ of tuples $(\hat{s}_t, \hat{a}_t, \hat{r}_t, \hat{s}_{t+1})$.

**IQL Training with $\mathcal{D}_{\text{gen}}$:** For offline policy learning stage, we employ IQL (Kostrikov et al., 2022) trained on the aggregated dataset $\mathcal{D}_{\text{gen}} \cup \mathcal{D}_{\text{tar}}$. For notational simplicity in the loss definitions that follow, we denote a generic transition from this combined set as $(s_t, a_t, r_t, s_{t+1})$, irrespective of its origin from $\mathcal{D}_{\text{gen}}$ or $\mathcal{D}_{\text{tar}}$. The network parameters for the Value function and Q-function are denoted by $\varphi$ (with $\varphi'$ for the target network and $\varphi^-$ for the stop gradient), and policy network parameters are denoted by $\omega$. The value function $V_\varphi$ is then trained using expectile regression:

$$\mathcal{L}_V(\varphi) = \mathbb{E}_{(s_t, a_t) \sim \mathcal{D}_{\text{gen}} \cup \mathcal{D}_{\text{tar}}} \left[ L_2^{\tau_V} \big( Q_{\varphi'}(s_t, a_t) - V_\varphi(s_t) \big) \right], \tag{B.12}$$

where $L_2^{\tau_V}(u) = \left| \tau_V - \mathbb{1}[u < 0] \right| u^2$ is the asymmetric $L_2$ loss, $\mathbb{1}(\cdot)$ is the indicator function, and $\tau_V \in (0, 1)$ controls the degree of conservatism, which is fixed to $\tau_V = 0.7$ for all environments in this work. In addition, the Q-function $Q_\varphi$ is updated via a standard TD error minimization:

$$\mathcal{L}_Q(\varphi) = \mathbb{E}_{(s_t, a_t, r_t, s_{t+1}) \sim \mathcal{D}_{\text{gen}} \cup \mathcal{D}_{\text{tar}}} \left[ \left( r_t + \gamma V_{\varphi^-}(s_{t+1}) - Q_\varphi(s_t, a_t) \right)^2 \right]. \tag{B.13}$$

Finally, the policy $\pi_\omega$ is trained by maximizing an advantage-weighted log-likelihood, augmented with a behavior-regularization term:

$$\mathcal{L}_\pi(\omega) = \mathbb{E}_{(s_t, a_t) \sim \mathcal{D}_{\text{gen}} \cup \mathcal{D}_{\text{tar}}} \left[ \exp\left( \beta_{\text{Adv}} \, \text{Adv}(s_t, a_t) \right) \, \log \pi_\omega(a_t | s_t) \right]$$
$$+ \beta_{\text{reg}} \, \mathbb{E}_{s_t \sim \mathcal{D}_{\text{tar}}} \left[ D_{\text{KL}}(\hat{\pi}_b(\cdot | s_t) \, \| \, \pi_\omega(\cdot | s_t)) \right], \tag{B.14}$$

where $\text{Adv}(s_t, a_t) = Q_\varphi(s_t, a_t) - V_\varphi(s_t)$ is the advantage, weighted by a temperature parameter $\beta_{\text{Adv}}$, which is fixed to $\beta_{\text{Adv}} = 3$ in all environments in this work. The term $\hat{\pi}_b$ is the empirical behavior policy cloned from $\mathcal{D}_{\text{tar}}$ using a CVAE, and $\beta_{\text{reg}}$ is the hyperparameter controlling the strength of the KL-divergence penalty. The target Q-network $\varphi'$ is updated via an exponential moving average of $\varphi$.

## B.3 Network Architecture and Configurations

We employ four core neural network architectures to model state scores, transition scores, inverse dynamics, and reward estimation. These networks are designed with modular subcomponents to promote parameter sharing and maintain architectural consistency.

Mixture-based state score network($q_\theta^{\mathrm{mix}}$) estimates the score of perturbed states conditioned on diffusion time and a mixture label. The input consists of concatenated state $s_t$, time embedding via TimeMLP $\tau$, and label embedding via LabelMLP $y$. These inputs pass through an initial dense layer, followed by four residual blocks composed of dense layers with SiLU activations and skip connections, culminating in a linear projection back to $\mathbb{R}^{d_s}$, where $d_s$ is the state dimension. This architecture facilitates expanding the coverage of target states by modulating the label input during the sampling process.

Target-transition score network($q_\theta^{\mathrm{tran}}$) predicts the score of perturbed next states $s_{t+1}$, conditioned on the current state $s_t$ and diffusion time $\tau$. Inputs $s_{t+1}$, the time embedding TimeMLP$(\tau)$, and state embedding StateMLP$(s_t)$ are concatenated and processed identically via a dense layer, four residual SiLU blocks with skip connections, and a final linear projection to $\mathbb{R}^{d_s}$. This ensures the generated transitions align closely with the target domain dynamics.

Inverse dynamics model($\mathrm{Inv}_\psi$) maps concatenated state pairs $[s_t, s_{t+1}]$ to the action space $\mathbb{R}^{d_a}$. The network starts with a dense projection layer, followed by three residual MLP blocks each comprising layer normalization, dense layers with SiLU activations, and stochastic depth via DropPath. A final linear layer outputs the action vector.

Reward model($R_\psi$) shares the architectural backbone with the inverse dynamics model, regressing scalar reward values from state pairs $[s_t, s_{t+1}]$. It utilizes the same input projection and residual blocks but concludes with a single-unit linear output for the reward prediction.

Table B.1 organizes these architectures by listing inputs, layer compositions, and output specifications. To reduce redundancy, reusable modules such as TimeMLP, LabelMLP, StateMLP, and residual blocks are summarized separately in Table B.2. Details of environment-specific state and action dimensions ($d_s$ and $d_a$) are provided in Appendix C.3.

| Network | Layers |
|---|---|
| Mixture-based state score network ($q_\theta^{\mathrm{mix}}$) | Input: state $s_t$, time $\tau$, label $y$
Concat[$s_t$, TimeMLP($\tau$), LabelMLP(y)]
Dense(256)
Residual block (Dense(256, SiLU), skip) $\times 4$
Dense($d_s$) |
| Target-transition score network ($q_\theta^{\mathrm{tran}}$) | Input: next state $s_{t+1}$, time $\tau$, state $s_t$
Concat[$s_{t+1}$, TimeMLP($\tau$), StateMLP($s_t$)]
Dense(256)
Residual block (Dense(256, SiLU), skip) $\times 4$
Dense($d_s$) |
| Inverse dynamics model ($\mathrm{Inv}_\psi$) | Input: state $s_t$, next state $s_{t+1}$
Dense(256)
RB-ResMLP $\times 3$
Dense($d_a$) |
| Reward model ($R_\psi$) | Input: state $s_t$, next state $s_{t+1}$
Dense(256)
RB-ResMLP $\times 3$
Dense(1) |

Table B.1: Architectural specifications of the networks. Notation: $d_s$ denotes the state dimension and $d_a$ denotes the action dimension. Layer notation uses "Layer(Dim, Activate)": Dim is the output width (number of units), and Activate is the activation function; if Activate is omitted, the layer is a linear projection. "Block $\times K$" means the preceding block is repeated $K$ times.

| Component | Definition |
|-----------|------------|
| TimeMLP | Dense(1,128), SiLU; Dense(128,128), SiLU |
| LabelMLP | Dense(1,128), SiLU; Dense(128,128), SiLU |
| StateMLP | Dense($d_s$,128), SiLU; Dense(128,128), SiLU |
| RB-ResMLP | Dense(256,256), SiLU; Residual connection |

Table B.2: Definitions of reusable components.

## C  DETAILED EXPERIMENTAL SETUP

This section outlines the experimental setup in detail. It first summarizes the baselines used for comparison in section C.1, then describes the environment and offline dataset configurations and domain shifts considered in section C.2, and finally presents the hyperparameter choices for model training and algorithm parameters in section C.3.

### C.1  BASELINES EXPLANATION

This section summarizes the baseline algorithms used for comparison with TCE approach.

**IQL** (Kostrikov et al., 2022) is a widely used offline RL method that learns policies strictly within the support of the dataset, avoiding extrapolation to out-of-distribution samples. While stable, this design makes it difficult to learn meaningful policies when only limited target-domain data is available. The variant IQL* leverages both source and target datasets to stabilize training and expand state coverage. **Official Code:** https://github.com/ikostrikov/implicit_q_learning.

**DARA** (Liu et al., 2022) mitigates the effect of dynamics mismatch by training domain classifiers on state-action-next-state and state-action pairs to quantify domain discrepancy. This discrepancy is used to adjust source rewards, encouraging source data to better align with target dynamics. For this method, we follow the implementation provided in Lyu et al. (2025).

**BOSA** (Liu et al., 2024) introduces supported value estimation to constrain critic updates to plausible transitions under target dynamics. The actor updates only consider supported actions, preventing exploitation of unsupported out-of-distribution transitions. For this method, we follow the implementation provided in Lyu et al. (2025).

**SRPO** (Xue et al., 2023) constrains the learned policy distribution to remain close to the target state distribution by incorporating a KL divergence budget, inducing a reward shaping term via a domain discriminator to enforce consistency with target dynamics. For this method, we follow the implementation provided in Lyu et al. (2025).

**IGDF** (Wen et al., 2024) learns cross-domain contrastive representations that distinguish source from target transitions. The resulting scores filter source data during critic training to ensure only reliable source transitions contribute. **Official Code:** https://github.com/BattleWen/IGDF.

**OTDF** (Lyu et al., 2025) applies optimal transport to align source and target transitions by computing deviation scores and selectively weighting source samples. The policy update also integrates CVAE-based support regularization to ensure the learned policy remains consistent with the target action space. **Official Code:** https://github.com/dmksjfl/OTDF.

### C.2  ENVIRONMENTAL SETUP

This section details the experimental environments used to evaluate our approach. We adopt the cross-domain continuous control setup, including environments, dataset compositions, and domain shift configurations, proposed by Lyu et al. (2025). The benchmark is based on MuJoCo environments (Todorov et al., 2012) and features four agent types: HalfCheetah, Hopper, Walker2d, and Ant.

**Offline Datasets:**  The offline datasets consist of pre-collected data from both source and target domains, primarily drawn from the D4RL benchmark (Fu et al., 2020) and supplemented by the aforementioned setup.

Source domain data comprises three quality levels: `medium`, `medium-replay`, and `medium-expert`. The `medium` dataset contains 1M samples generated by a partially trained SAC policy. The `medium-replay` dataset includes all samples in the SAC replay buffer up to the point of medium performance (approx. 0.2M to 0.4M samples), thus mixing low- to medium-quality experiences. Finally, the `medium-expert` datasets combine 50% expert and 50% suboptimal data, with total sizes ranging from 1M to 2M transitions.

Target domain is designed to assess policy adaptation under significant distributional shifts. Three types of dynamics changes are introduced to the MuJoCo agents: **Morphology** shift(modifying the agent's physical structure), **Kinematic** shift(restricting joint rotations to simulate malfunctions), and **Gravity** Shift(altering gravitational acceleration). To reflect realistic data scarcity, the target datasets are limited to a small number of samples (typically under 5K per dataset) and are provided across `medium`, `medium-expert`, and `expert` quality levels. Table C.1 summarizes the dataset sizes by domain and quality.

| | **Source Domain** | | **Target Domain :** | Morphology | Kinematic | Gravity |
|---|---|---|---|---|---|---|
| | medium | 1M | medium | 5K | 5K | 5K |
| HalfCheetah | medium-replay | 0.2M | medium-expert | 5K | 5K | 5K |
| | medium-expert | 2M | expert | 5K | 5K | 5K |
| | medium | 1M | medium | 5K | 5K | 5K |
| Hopper | medium-replay | 0.4M | medium-expert | 4.3K | 5K | 4.3K |
| | medium-expert | 2M | expert | 5K | 5K | 5K |
| | medium | 1M | medium | 5K | 5K | 5K |
| Walker2d | medium-replay | 0.3M | medium-expert | 3.5K | 4.4K | 4.8K |
| | medium-expert | 2M | expert | 5K | 5K | 5K |
| | medium | 1M | medium | 5K | 5K | 5K |
| Ant | medium-replay | 0.3M | medium-expert | 5K | 5K | 3.1K |
| | medium-expert | 2M | expert | 5K | 5K | 5K |

Table C.1: Dataset sizes by domain and data quality

**Evaluation Metric:** All results are presented as normalized scores to fairly compare across environments with varying return scales:

$$NS = \frac{J - J_r}{J_e - J_r} \times 100,$$

where $J$ is the return of the evaluated policy, and $J_r$ and $J_e$ represent returns from random and expert policies in the target domain, respectively. Reference scores proposed by Lyu et al. (2025) for each agent and target domain scenario are shown in Table C.2.

| Agent Type | Domain Shifts | Reference min score $J_r$ | Reference max score $J_e$ |
|---|---|---|---|
| HalfCheetah | Morphology | -280.18 | 9713.59 |
| HalfCheetah | Kinematic | -280.18 | 7065.03 |
| HalfCheetah | Gravity | -280.18 | 9509.15 |
| Hopper | Morphology | -26.34 | 3152.75 |
| Hopper | Kinematic | -26.34 | 2842.73 |
| Hopper | Gravity | -26.34 | 3234.3 |
| Walker2d | Morphology | 10.8 | 4398.43 |
| Walker2d | Kinematic | 10.8 | 3257.51 |
| Walker2d | Gravity | 10.8 | 5154.71 |
| Ant | Morphology | -325.6 | 5722.01 |
| Ant | Kinematic | -325.6 | 5122.57 |
| Ant | Gravity | -325.6 | 4317.07 |

Table C.2: Reference minimum score $J_r$ and maximum score $J_e$ by agent and domain shifts

| Agent Type | State Dimension($d_s$) | Action Dimension($d_a$) |
|:---:|:---:|:---:|
| HalfCheetah | 17 | 6 |
| Hopper | 11 | 3 |
| Walker2d | 17 | 6 |
| Ant | 111 | 8 |

Table C.3: state dimension($d_s$) and action dimension($d_a$) of each agent.

**Domain Shifts:**   To simulate realistic but significant distributional differences between source and target domains, we apply three primary types of domain shifts in the MuJoCo environments: Morphology, Kinematics, and Gravity. These shifts directly impact the agent's control dynamics and pose challenges to policy generalization.

Morphology shift involves structural modifications to the agent's body by editing MuJoCo XML files to change limb sizes and dimensions of body parts. These changes alter movement capabilities without fundamentally breaking agent functionality. For example, in HalfCheetah, the thigh lengths are significantly reduced, requiring the agent to adapt its gait to control shorter limbs efficiently. In Hopper, the head size is increased from 0.05 to 0.125, representing a 60% expansion in torso diameter. In Walker2d, the right leg is elongated by increasing the lengths of the thigh, leg, and foot segments through geometry modifications to their endpoints. In Ant, the sizes of the front ankle capsules are reduced, resulting in smaller front feet compared to the default configuration. These alterations create physical domain gaps that affect locomotion efficiency and balance.

Kinematic shift emulates impairments or restrictions in joint mobility by significantly narrowing the allowed range of joint rotations. This effectively simulates partial joint failure or stiffened joints in the target domain. Specifically, in HalfCheetah, the back thigh joint's rotational freedom is drastically reduced by approximately 99%. For Hopper, the head and foot joint ranges are contracted by roughly 99% and 60%, respectively. In Walker2d, the right foot joint experiences a 71% decrease in allowable rotation, while for Ant, the hip joints of the front legs are tightened by 43%. These modifications limit the agents' maneuverability and adaptability in the altered domain.

Gravity shift reduces the gravitational force magnitude acting on the agents by halving the gravity parameter in the target domain compared to the source. The source domain uses standard Earth gravity of -9.81 m/s$^2$, while the target domain gravity is set to -4.905 m/s$^2$, exactly half. This results in a lighter environment where decreased downward force alters the agents' balance, contact dynamics, and locomotion. Such a change demands the learned policies adapt to modified physical interactions and energy requirements.

All these domain shifts are implemented through precise modifications in the MuJoCo XML configuration files following the framework of Lyu et al. (2025), enabling a comprehensive evaluation of cross-domain offline reinforcement learning under varied and challenging environment changes.

HalfCheetah morphology shift : shortened thigh capsules

```
<geom fromto="0 0 0 -0.0001 0 -0.0001" name="bthigh" size="0.046" type="
    capsule"/>
<body name="bshin" pos="-0.0001 0 -0.0001">
<geom fromto="0 0 0 0.0001 0 0.0001" name="fthigh" size="0.046" type="
    capsule"/>
<body name="fshin" pos="0.0001 0 0.0001">
```

Hopper morphology shift : reduced torso size

```
<geom friction="0.9" fromto="0 0 1.45 0 0 1.05" name="torso_geom" size="
    0.125" type="capsule"/>
```

Walker2d morphology shift : shortened thigh, elongated leg

```
<body name="thigh" pos="0 0 1.05">
  <joint axis="0 -1 0" name="thigh_joint" pos="0 0 1.05" range="-150 0"
      type="hinge"/>
```

```
<geom friction="0.9" fromto="0 0 1.05 0 0 1.045" name="thigh_geom" size
    ="0.05" type="capsule"/>
<body name="leg" pos="0 0 0.35">
  <joint axis="0 -1 0" name="leg_joint" pos="0 0 1.045" range="-150 0"
      type="hinge"/>
  <geom friction="0.9" fromto="0 0 1.045 0 0 0.3" name="leg_geom" size=
      "0.04" type="capsule"/>
  <body name="foot" pos="0.2 0 0">
    <joint axis="0 -1 0" name="foot_joint" pos="0 0 0.3" range="-45 45"
        type="hinge"/>
    <geom friction="0.9" fromto="-0.0 0 0.3 0.2 0 0.3" name="foot_geom"
        size="0.06" type="capsule"/>
  </body>
</body>
</body>
```

Ant morphology shift : smaller front ankle capsules

```
<geom fromto="0.0 0.0 0.0 0.1 0.1 0.0" name="left_ankle_geom" size="0.08"
    type="capsule"/>
<geom fromto="0.0 0.0 0.0 -0.1 0.1 0.0" name="right_ankle_geom" size="
    0.08" type="capsule"/>
```

HalfCheetah kinematic shift : restricted back thigh joint

```
<joint axis="0 1 0" damping="6" name="bthigh" pos="0 0 0" range="-.0052
    .0105" stiffness="240" type="hinge"/>
```

Hopper kinematic shift : narrowed head and foot joints

```
<joint axis="0 -1 0" name="thigh_joint" pos="0 0 1.05" range="-0.15 0"
    type="hinge"/>
<joint axis="0 -1 0" name="foot_joint" pos="0 0 0.1" range="-18 18" type=
    "hinge"/>
```

Walker2d kinematic shift : constrained foot joint

```
<joint axis="0 -1 0" name="foot_joint" pos="0 0 0.1" range="-0.45 0.45"
    type="hinge"/>
```

Ant kinematic shift : limited hip joints

```
<joint axis="0 0 1" name="hip_1" pos="0.0 0.0 0.0" range="-0.3 0.3" type=
    "hinge"/>
<joint axis="0 0 1" name="hip_2" pos="0.0 0.0 0.0" range="-0.3 0.3" type=
    "hinge"/>
```

Gravity shift : half of the original gravity scale

```
<option gravity="0 0 -4.905" timestep="0.01"/>
```

## C.3 HYPERPARAMETER SETUP

We summarize the hyperparameters related to model training, data generation, and reinforcement learning in Table C.4, and algorithm-specific hyperparameters in Table C.5.

For the HalfCheetah morphology shifts, setting the maximum label bound $y_{\max}$ to 0.1 yields better performance, while in other environments it is fixed at 0.2. The Z-score filtering threshold $z_{\text{th}}$ is set to a more stringent value of 2.5 for Walker2d morphology shifts with transfer to a `medium-expert` target dataset and for Walker2d kinematic shifts, whereas $z_{\text{th}} = 3.0$ is used for other environments. The policy regularization coefficient $\beta_{\text{reg}}$ is increased to 2.0 for Walker2d morphology shifts in `medium-replay-to-medium-expert` and `medium-expert-to-medium-expert` settings, with a value of 0.001 adopted for other environments. For TCE(NN), the mixture coefficient $\lambda$ is set to 0.9 in HalfCheetah gravity shifts, and 0.1 for other environments.

| | | Name | Value |
|---|---|---|---|
| Shared Hyper-parameters | Model training | Learning rate for $q_\theta^{\text{mix}}$, $q_\theta^{\text{tran}}$ | 1e-4 |
| | | Optimizer for $q_\theta^{\text{mix}}$, $q_\theta^{\text{tran}}$ | Adam |
| | | Batch size for $q_\theta^{\text{mix}}$, $q_\theta^{\text{tran}}$ | 128 |
| | | Training epochs for $q_\theta^{\text{mix}}$ | 10K |
| | | Training epochs for $q_\theta^{\text{tran}}$ | 5K |
| | | Learning rate for $\text{Inv}_\psi$, $R_\psi$ | 1e-3 |
| | | Optimizer for $\text{Inv}_\psi$, $R_\psi$ | Adam |
| | | Batch size for $\text{Inv}_\psi$, $R_\psi$ | 128 |
| | | Training epochs for $\text{Inv}_\psi$, $R_\psi$ | 1K |
| | | Noise schedule $\alpha_{\min}$ | 0.1 |
| | | Noise schedule $\alpha_{\max}$ | 20 |
| | Sampling | Denoising steps $K$ | 0.5K |
| | RL training | Learning rate for Actor, Critic | 3e-4 |
| | | Optimizer for Actor, Critic | Adam |
| | | Batch size for target sample | 128 |
| | | Batch size for generated sample | 128 |
| | | Training steps | 1M |

Table C.4: Shared hyperparameters

| | Name | Value |
|---|---|---|
| Hyperparameter Setup | $y_{\max}$ | 0.1 for HalfCheetah-morph
0.2 for other environments |
| | $z_{\text{th}}$ | 2.5 for Walker2d-morph {m, m-r, m-e}-to-m-e
3.0 for other environments |
| | $\beta_{\text{reg}}$ | 2.0 for Walker2d-morph {m-r, m-e}-to-m-e
0.5 for Walker2d-kinematic
0.001 for other environments |
| | $\lambda$ | 0.9 for HalfCheetah-gravity
0.1 for other environments |

Table C.5: Algorithm-specific hyperparameters. The -morph suffix denotes morphology shifts. Dataset abbreviations are as follows: m for `medium`, m-r for `medium-replay`, and m-e for `medium-expert`. The notation {A, B}-to-C denotes transfer from a source dataset of quality A or B to a target dataset of quality C.

## D ADDTIONAL PERFORMANCE COMPARISON

In Section D.1, we present performance comparisons in gravity shift environments not covered in the main paper. Section D.2 evaluates TCE under larger domain gaps using high-gravity settings from the ODRL benchmark (Lyu et al., 2024b). Section D.3 reports comparative results against Meta-DT (Wang et al., 2024) and recently proposed DmC (Le Pham Van et al., 2025) method.

### D.1 GRAVITY SHIFT

Table D.1 reports the results for gravity shifts. While all TCE approaches outperform baselines across most tasks, TCE(NN) achieves the highest average returns. TCE(OG) shows lower performance in HalfCheetah tasks, likely due to the very low quality of target data, which hinders reliable transition learning. Instead, TCE(NN) adopts a higher mixing ratio ($\lambda = 0.9$) for HalfCheetah gravity shifts to better balance generated and target samples, thereby improving scores in these challenging domains. Despite minor tuning, TCE(NN) and TCE(OG) both exhibit strong results, confirming their robust effectiveness for gravity shift adaptation.

| Src. | Tgt. | IQL* | DARA | BOSA | SRPO | IGDF | OTDF | TCE(OG) | TCE(NN) |
|------|------|------|------|------|------|------|------|---------|---------|
| half-m | m | 39.6±3.3 | **41.2±3.9** | 38.9±4.0 | 36.9±4.5 | 36.6±5.5 | 40.7±7.7 | 15.9±0.8 | 39.6±1.3 |
| half-m | m-e | 39.6±3.7 | 40.7±2.8 | 40.4±3.0 | 40.7±2.3 | 38.7±6.2 | 28.6±3.2 | 5.5±0.6 | **41.8±3.3** |
| half-m | e | 42.4±3.8 | 39.8±4.4 | 40.5±3.9 | 39.4±1.6 | 39.6±4.6 | 36.1±5.3 | 11.6±2.63 | **44.7±2.3** |
| half-m-r | m | 20.1±5.0 | 17.6±6.2 | 20.0±4.9 | 17.5±5.2 | 14.4±2.2 | **21.5±6.5** | 5.2±1.3 | 18.2±0.3 |
| half-m-r | m-e | 17.2±1.6 | **20.2±5.2** | 16.7±4.2 | 16.3±1.7 | 10.0±2.5 | 14.7±4.1 | 5.6±1.0 | 19.8±1.3 |
| half-m-r | e | 20.7±5.5 | 22.4±1.7 | 15.4±4.2 | **23.1±4.0** | 15.3±3.7 | 11.4±1.9 | 21.8±1.6 | 16.7±2.4 |
| half-m-e | m | 38.6±6.0 | 37.8±3.3 | 41.8±5.1 | **42.5±2.3** | 37.7±7.3 | 39.5±3.5 | 9.1±0.3 | 41.1±2.1 |
| half-m-e | m-e | 39.6±3.0 | 39.4±4.4 | 38.7±2.7 | **43.3±2.7** | 40.7±3.2 | 32.4±5.5 | 10.4±0.7 | 42.9±0.3 |
| half-m-e | e | 43.4±0.9 | **45.3±1.3** | 39.9±2.7 | 43.3±3.0 | 41.1±4.1 | 26.5±9.1 | 42.5±0.12 | 44.9±0.2 |
| hopp-m | m | 11.2±1.1 | 17.3±3.8 | 15.2±3.3 | 12.4±1.0 | 15.3±3.5 | 32.4±8.0 | **58.7±4.5** | 53.6±1.3 |
| hopp-m | m-e | 14.7±3.6 | 15.4±2.5 | 21.1±9.3 | 14.2±1.8 | 15.1±3.6 | 24.2±3.6 | **51.6±6.3** | 42.0±4.8 |
| hopp-m | e | 12.5±1.6 | 19.3±10.5 | 12.7±1.7 | 11.8±0.9 | 14.4±0.8 | 33.7±7.8 | **38.9±10.8** | 37.5±2.3 |
| hopp-m-r | m | 13.9±2.9 | 10.7±4.3 | 3.3±1.9 | 14.0±2.6 | 15.3±4.4 | 31.1±13.4 | **61.8±4.2** | 52.9±1.1 |
| hopp-m-r | m-e | 13.3±6.3 | 12.5±5.6 | 4.6±1.7 | 14.4±4.2 | 15.4±5.5 | 24.2±6.1 | 41.9±10.4 | **46.5±5.1** |
| hopp-m-r | e | 11.0±2.6 | 14.3±6.0 | 3.2±0.8 | 16.4±5.0 | 16.1±4.0 | 31.0±9.8 | **39.5±8.1** | 34.1±3.0 |
| hopp-m-e | m | 19.1±6.6 | 18.5±12.3 | 15.9±5.9 | 19.7±8.5 | 22.3±5.4 | 26.4±10.1 | **54.7±5.1** | 49.8±4.2 |
| hopp-m-e | m-e | 16.8±2.7 | 16.0±6.1 | 17.3±2.5 | 15.8±3.3 | 16.6±7.7 | 28.3±6.7 | **45.2±8.9** | 44.3±2.3 |
| hopp-m-e | e | 20.9±4.1 | 23.9±14.8 | 23.2±7.9 | 21.4±1.9 | 26.0±9.2 | **44.9±10.6** | 36.8±5.8 | 32.3±5.3 |
| walk-m | m | 28.1±12.9 | 28.4±13.7 | 38.0±11.2 | 21.4±7.0 | 22.1±8.4 | 36.6±2.3 | 38.3±5.1 | **40.4±6.5** |
| walk-m | m-e | 35.7±4.7 | 30.7±9.7 | 40.9±7.2 | 34.0±9.9 | 35.4±9.1 | **44.8±7.5** | 21.8±3.7 | 41.9±5.5 |
| walk-m | e | 37.3±8.0 | 36.0±7.0 | 41.3±8.6 | 39.5±3.8 | 36.2±13.6 | 44.0±4.0 | 26.7±4.5 | **47.8±7.3** |
| walk-m-r | m | 14.6±2.5 | 14.1±6.1 | 7.6±5.8 | 17.9±3.8 | 11.6±4.6 | 32.7±7.0 | **38.9±6.2** | 37.0±7.1 |
| walk-m-r | m-e | 15.3±1.9 | 15.9±5.8 | 4.8±5.8 | 15.3±4.5 | 13.9±6.5 | **31.6±6.1** | 18.8±3.3 | 17.5±4.3 |
| walk-m-r | e | 15.8±7.2 | 15.7±4.5 | 7.1±4.6 | 13.7±8.1 | 15.2±5.3 | 31.3±5.3 | 31.3±9.6 | **45.0±5.7** |
| walk-m-e | m | 39.9±13.1 | 41.6±13.0 | 32.3±7.2 | **46.4±3.5** | 33.8±3.1 | 30.2±9.8 | 36.3±3.6 | 46.2±2.1 |
| walk-m-e | m-e | 49.1±6.9 | 45.8±9.4 | 40.1±4.5 | 36.4±3.4 | 44.7±2.9 | **53.3±7.1** | 47.3±6.5 | 50.1±7.6 |
| walk-m-e | e | 40.4±11.9 | 56.4±3.5 | 43.7±4.4 | 45.8±8.0 | 45.3±10.4 | **61.1±3.4** | 26.7±2.9 | 43.8±8.3 |
| ant-m | m | 10.2±1.8 | 9.4±0.9 | 12.4±2.0 | 11.7±1.0 | 11.3±1.3 | 45.1±12.4 | 52.1±2.4 | **58.3±3.5** |
| ant-m | m-e | 9.4±1.2 | 10.0±0.9 | 11.6±1.3 | 10.2±1.2 | 9.4±1.4 | 33.9±5.4 | **42.7±4.7** | 36.9±4.8 |
| ant-m | e | 10.2±0.3 | 9.8±0.6 | 11.8±0.4 | 9.5±0.6 | 9.7±1.6 | 33.2±9.0 | **52.6±6.5** | 46.2±13.1 |
| ant-m-r | m | 18.9±2.6 | 21.7±2.1 | 13.9±1.5 | 18.7±1.7 | 19.6±1.0 | 29.6±10.7 | **55.5±3.2** | 51.3±3.7 |
| ant-m-r | m-e | 19.1±3.0 | 18.3±2.1 | 15.9±2.7 | 18.7±1.8 | 20.3±1.6 | 25.4±2.1 | 40.1±4.5 | 36.1±8.9 |
| ant-m-r | e | 18.5±0.9 | 20.0±1.3 | 14.5±1.7 | 19.9±2.1 | 18.8±2.1 | 24.5±2.8 | **53.2±4.9** | 53.1±3.3 |
| ant-m-e | m | 9.8±2.4 | 8.1±1.8 | 8.1±3.0 | 8.4±2.1 | 8.9±1.5 | 18.6±11.9 | **54.4±1.6** | 52.0±2.1 |
| ant-m-e | m-e | 9.0±0.8 | 6.4±1.4 | 6.2±1.5 | 6.1±3.5 | 7.2±2.9 | 34.0±9.4 | **44.7±4.5** | 35.3±2.4 |
| ant-m-e | e | 9.1±2.6 | 10.4±2.9 | 4.2±3.9 | 8.8±1.0 | 9.2±1.5 | 23.2±2.9 | **52.9±4.1** | 44.4±3.1 |
| **Total Score** | | 825.0 | 851.0 | 763.2 | 825.5 | 803.6 | 1160.7 | 1291 | **1486.0** |

Table D.1: Performance comparison under gravity shifts. Abbrev.: half=HalfCheetah, hopp=Hopper, walk=Walker2d, ant=Ant; m=medium, m-r=medium-replay, e=expert, m-e=medium-expert. "Src./Tgt." denote source/target dataset qualities of the two domains. Numbers are mean±std over 5 seeds from normalized scores; best per row in **bold**, tcbsecond best in underbar.

## D.2 GRAVITY SHIFT UNDER ODRL BENCHMARK

To evaluate our method under an extreme domain gap, we conduct additional experiments on the Ant task from ODRL (Lyu et al., 2024b), where gravity is increased by a factor of 5 (from standard $-9.81$ m/s$^2$ to $-49.05$ m/s$^2$). We compare TCE(OG) against IQL* and OTDF under identical settings, maintaining the hyperparameters $y_{\max} = 0.2$ and $z_{\mathrm{th}} = 3$. As shown in Table D.2, TCE(OG) demonstrates substantially superior performance over the baselines. This result reinforces our main claim that TCE is highly robust even in environments with severe dynamic shifts.

| Src. | Tgt. | IQL* | OTDF | TCE(OG) |
|------|------|------|------|---------|
| ant-m | m | 31.9±0.2 | 34.5±0.2 | **70.8±0.5** |
| ant-m | e | 31.3±0.3 | 38.2±3.1 | **86.9±0.8** |
| ant-m-r | m | 18.6±0.2 | 24.8±0.4 | **44.3±1.6** |
| ant-m-r | e | 18.6±0.1 | 23.1±1.2 | **70.9±5.5** |
| ant-m-e | m | 30.1±0.0 | 35.0±0.9 | **70.1±1.2** |
| ant-m-e | e | 31.6±0.1 | 33.7±1.1 | **82.5±0.4** |
| **Total Score** | | 162.1 | 189.3 | **425.5** |

Table D.2: Performance comparison on gravity-shift(5.0) tasks.

### D.3 Additional Performance Comparison under Kinematic Shifts

In this section, we compare our two TCE variants, TCE (OG) and TCE (NN), against two additional baselines: Meta-DT* (Wang et al., 2024), which performs offline meta-learning with a Decision Transformer architecture on offline data collected from multiple tasks to improve task generalization, and DmC (Le Pham Van et al., 2025), which selects source transitions that are closest to the target data and uses diffusion to generate additional source-like transitions around them. Since Meta-DT is not originally designed for a cross-domain setup, we follow the IQL protocol and train it on a union of source and target data, and refer to this variant as Meta-DT*. These comparisons clarify how the cross-domain offline RL setting considered in this work differs from the conventional meta offline setup and how our approach differs from existing methods that explicitly generate source-like transitions.

| Src. | Tgt. | IQL* | Meta-DT* | OTDF | DmC | TCE(OG) | TCE(NN) |
|------|------|------|----------|------|-----|---------|---------|
| half-m | m | 12.3±1.2 | 13.4 ± 3.4 | 40.2±0.0 | 38.5±1.4 | **41.9±0.9** | 41.4±0.1 |
| half-m | m-e | 10.8±1.9 | 8.5±0.6 | 10.1±4.0 | 19.1±1.0 | 39.7±1.1 | **40.5±0.5** |
| half-m | e | 12.6±1.7 | 5.0±0.1 | 8.7±2.0 | **13.1±0.8** | 11.9±4.6 | 7.5±1.1 |
| half-m-r | m | 10.0±5.4 | 5.5±0.7 | 37.8±2.1 | 19.5±1.8 | **41.8±0.5** | 40.2±0.9 |
| half-m-r | m-e | 6.5±3.1 | 7.5±1.1 | 9.7±2.0 | 11.4±2.1 | **40.8±1.4** | 33.6±6.4 |
| half-m-r | e | 13.6±1.4 | 6.4±2.8 | 7.2±1.4 | **15.6±2.9** | 15.2±6.4 | 2.9±0.1 |
| half-m-e | m | 21.8±6.5 | 4.7±0.2 | 30.7±9.6 | 38.4±1.4 | **42.0±0.2** | 41.1±0.5 |
| half-m-e | m-e | 7.6±1.4 | 7.7±3.2 | 10.9±4.2 | 24.1±4.6 | **41.2±0.6** | 35.8±1.8 |
| half-m-e | e | 9.1±2.4 | 2.8±0.2 | 3.2±0.6 | **13.4±2.0** | 9.5±7.4 | 7.5±1.6 |
| hopp-m | m | 58.7±8.4 | 5.0±0.2 | 65.6±1.9 | **69.8±2.3** | 66.8±0.5 | 66.3±0.2 |
| hopp-m | m-e | 68.5±12.4 | 4.9±0.2 | 55.4±25.1 | **78.2±5.1** | 72.1±4.1 | 67.3±2.9 |
| hopp-m | e | 79.9±35.5 | 5.2 ± 0.2 | 35.0±19.4 | 59.8±21.8 | **91.5±6.3** | 78.2±17.6 |
| hopp-m-r | m | 36.0±0.1 | 36.4± 0.2 | 35.5±12.2 | 64.8±2.4 | 65.1±0.9 | **66.2±0.2** |
| hopp-m-r | m-e | 36.1±0.1 | 36.3± 0.1 | 47.5±14.6 | 69.7±7.5 | **72.0±3.7** | 63.9±14.3 |
| hopp-m-r | e | 36.0±0.1 | 37.6± 0.1 | 49.9±30.5 | 69.9±18.0 | **96.8±2.4** | 85.1±2.5 |
| hopp-m-e | m | 66.0±0.5 | 3.5± 0.4 | 65.3±2.4 | **69.6±1.3** | 66.6±0.6 | 66.2±0.2 |
| hopp-m-e | m-e | 45.1±15.7 | 8.5± 2.3 | 38.6±15.9 | 75.5±9.6 | **76.0±2.0** | 72.7±3.1 |
| hopp-m-e | e | 44.9±19.8 | 6.4±0.2 | 29.9±11.3 | 64.5±24.2 | 89.2±8.4 | **89.7±4.2** |
| walk-m | m | 34.3±9.8 | 5.0±0.3 | 49.6±18.0 | **63.2±4.2** | 60.4±1.9 | 54.1±2.1 |
| walk-m | m-e | 30.2±12.5 | 15.7±2.0 | 43.5±16.4 | **53.5±7.0** | 46.2±12.1 | 19.8±1.3 |
| walk-m | e | 56.4±18.2 | 10.0±0.9 | 46.7±13.6 | **70.5±12.0** | 59.3±4.2 | 33.4±1.5 |
| walk-m-r | m | 11.5±7.1 | 3.4±1.0 | 49.7±9.7 | **52.9±8.4** | 50.2±3.7 | 45.1±4.5 |
| walk-m-r | m-e | 9.7±3.8 | 14.6±0.1 | **55.9±17.1** | 36.4±5.4 | 37.1±11.8 | 21.3±5.5 |
| walk-m-r | e | 7.7±4.8 | 8.9±1.0 | 51.9±7.9 | 44.4±8.5 | **53.0±7.9** | 23.1±2.9 |
| walk-m-e | m | 41.8±8.8 | 8.5±0.8 | 44.6±6.0 | **59.4±6.8** | 55.2±2.5 | 54.9±3.2 |
| walk-m-e | m-e | 22.2±8.7 | 10.2±8.7 | 16.5±7.2 | **53.2±7.3** | 31.2±4.8 | 24.7±3.8 |
| walk-m-e | e | 26.3±10.4 | 5.7±2.6 | 42.4±9.1 | **69.2±7.0** | 47.1±18.1 | 25.2±6.1 |
| ant-m | m | 50.0±5.6 | 13.9±0.7 | 55.4±0.0 | **62.1±0.6** | 53.2±1.9 | 47.5±1.9 |
| ant-m | m-e | 57.8±7.2 | 14.8±0.3 | 60.7±3.6 | **68.9±1.0** | 61.4±2.0 | 64.2±9.5 |
| ant-m | e | 59.6±18.5 | 14.9±0.1 | 90.4±4.8 | 92.1±3.5 | 92.7±2.8 | **93.8±3.4** |
| ant-m-r | m | 43.7±4.6 | 22.3±0.2 | 52.8±4.4 | **61.9±0.5** | 54.6±1.4 | 51.0±3.2 |
| ant-m-r | m-e | 36.5±5.9 | 21.0±0.1 | 54.2±5.2 | 58.8±3.6 | 61.6±2.4 | **61.7±5.4** |
| ant-m-r | e | 24.4±4.8 | 26.8±2.2 | 74.7±10.5 | 43.8±2.6 | 92.0±2.4 | **94.2±0.2** |
| ant-m-e | m | 49.5±4.1 | 13.5±0.1 | 50.2±4.3 | **60.6±1.3** | 55.6±1.4 | 55.1±3.7 |
| ant-m-e | m-e | 37.2±2.0 | 13.6±0.1 | 48.8±2.7 | 60.4±3.7 | 59.1±3.4 | **62.1±0.2** |
| ant-m-e | e | 18.7±8.1 | 18.1±1.2 | 78.4±12.2 | 76.0±4.1 | **94.2±3.2** | 90.3±1.3 |
| **Total Score** | | 1193.0 | 446.2 | 1547.6 | 1902.2 | **2044.2** | 1828.1 |

Table D.3: Performance comparison of TCE with DmC and Meta-DT* under kinematic shifts

TCE still achieves substantially better performance than the additional baselines. Compared with Meta-DT, a representative meta offline RL method, the difference in objective is fundamental. Meta offline RL is designed to improve task generalization from offline data collected in multiple tasks, whereas cross-domain offline RL explicitly aims to enable transfer between two environments with different domains. Concretely, when Meta-DT* is trained on the offline data from both the source and target, it simply learns the source task from the source data and the target task from the target data. Note that to ensure a fair offline comparison, Meta-DT was evaluated in a zero-shot manner without any online interaction with the target domain. Under this constraint, it does not selectively identify and exploit source samples that are particularly useful for the target task. As a result, when the domain gap is large as in our cross-domain setting, meta offline RL methods such as Meta-DT* provide little benefit for learning the target policy and can even underperform IQL*, which directly uses both datasets to optimize the target policy.

DmC, on the other hand, is almost the opposite of our motivation, since it generates additional source-like transition samples. Although DmC can outperform TCE in a few environments, TCE consistently achieves better performance in most cases. Because cross-domain RL is ultimately concerned with learning a high-quality target policy, generating target-like transitions is more desirable, and Theorem 1 shows that such transitions reduce the domain gap more effectively. This explains why TCE tends to yield higher returns even when only a small amount of target data is available. There exist settings such as the gravity shift case where learning primarily from the source domain is more beneficial than exploiting the limited target data, in which methods like DmC can perform competitively or even better. Nevertheless, as observed empirically, TCE outperforms DmC on the majority of tasks.

## E    COMPUTATIONAL COMPLEXITY COMPARISONS

We compare the runtime and GPU memory usage of IQL* (Kostrikov et al., 2022), OTDF (Lyu et al., 2025), and our proposed TCE on morphology shift (`medium-to-expert`) tasks. All experiments were conducted on a server equipped with AMD EPYC 7513 32-Core CPUs and eight NVIDIA RTX 3090 GPUs running Ubuntu 20.04. Across all methods, the offline RL policy training phase is identical, requiring approximately three hours. OTDF incorporates an additional data selection step that takes roughly 18 minutes. In contrast, TCE involves training the score networks, which requires about two hours, followed by a transition sampling phase of approximately 10–12 minutes. These runtime costs are consistent across different agent types, remaining comparable even for the Ant task, which incurs only a marginal increase despite its larger state dimension. throughout the process, TCE maintains a GPU memory footprint of approximately 2 GB. As shown in Table E.1, while TCE introduces modest additional computation for score modeling, this cost is justified by the critical role of coverage expansion in achieving superior cross-domain offline RL performance in practice.

|                    | Model Training | Data Sampling | Offline RL | GPU Memory |
|--------------------|----------------|---------------|------------|------------|
| IQL*               | –              | –             | 3h         | –          |
| OTDF               | –              | 18 min        | 3h         | –          |
| TCE (HalfCheetah)  | 1h 55m         | 10m           | 3h         | 2 GB       |
| TCE (Hopper)       | 1h 55m         | 10m           | 3h         | 2 GB       |
| TCE (Walker2d)     | 1h 55m         | 10m           | 3h         | 2 GB       |
| TCE (Ant)          | 2h 5m          | 12m           | 3h 15m     | 2 GB       |

Table E.1: Comprehensive runtime and memory usage comparison for all methods in the morphology shift (`medium-to-expert`) setting.

# F    SAMPLE RELIABILITY ANALYSIS ON ADDITIONAL ENVIRONMENTS

To further demonstrate the generality and reliability of our method, we conduct sample reliability analysis on Hopper and Walker2d environments, which are not covered in the main paper, using the `medium-to-expert` setting.

**Hopper**    Fig. F.1(a) shows that outlier and training samples have similar NN-distance distributions, with minimal separation. In Fig. F.1(b), transition KL divergence increases moderately as $y_{\max}$ grows up to 0.2, but then rises sharply at higher $y_{\max}$ values; throughout, outlier samples consistently exhibit greater KL than generated samples. Fig. F.1(c) shows reward errors for outlier samples are always higher and increase rapidly for larger $y_{\max}$.

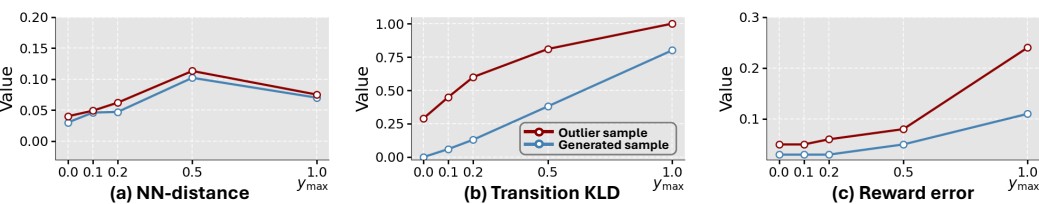

Figure F.1: Sample reliability with respect to $y_{\max}$ in Hopper morphology shifts. (a) NN-distance between generated states $\mathcal{D}_{\text{gen}}^{1-\lambda}$ and true states in $\mathcal{D}_{\text{tar}} \cup \mathcal{D}_{\text{src}}$. (b) Transition KL divergence(normalized) and (c) reward error between models trained on limited and sufficient target data.

**Walker2d**    Fig. F.2(a) shows that for small $y_{\max}$ there is minimal difference in NN-distance between outlier and training samples, whereas their separation becomes pronounced as $y_{\max}$ increases. In Fig. F.2(b), the transition KL divergence grows steadily with increasing $y_{\max}$, and outlier samples consistently exhibit much higher KL than training samples. Fig. F.2(c) further shows that reward prediction errors for outlier samples are always larger and grow rapidly at higher $y_{\max}$. Therefore, choosing an appropriate $y_{\max}$ and applying Z-score filtering are important to improve dataset reliability and downstream learning stability.

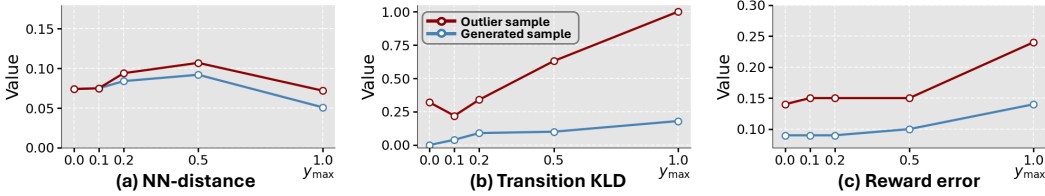

Figure F.2: Sample reliability with respect to $y_{\max}$ in Walker2d morphology shifts. (a) NN-distance between generated states $\mathcal{D}_{\text{gen}}^{1-\lambda}$ and true states in $\mathcal{D}_{\text{tar}} \cup \mathcal{D}_{\text{src}}$. (b) Transition KL divergence(normalized) and (b) reward error between models trained on limited and sufficient target data.

# G  ADDITIONAL ABLATION STUDY

This section presents additional ablation studies not covered in the main text. Section G.1 analyzes the sensitivity to the denoising step $K$, and Section G.2 compares TCE against a baseline that augments source state pairs with inverse target actions.

## G.1  ABLATION STUDY ON DENOISING STEP $K$

We examine the sensitivity of TCE to the denoising step hyperparameter $K$, which determines the discretization granularity of the reverse process. Table G.1 presents the results of TCE(OG) on the Ant morphology-shift environment across $K \in \{100, 200, 500\}$, using $K = 500$ as the default configuration. While $K = 500$ generally yields the highest average scores, the performance differences across varying $K$ values are minimal and mostly fall within the standard deviation ranges. This result demonstrates that TCE is not highly sensitive to the choice of $K$, ensuring robust performance without the need for precise hyperparameter tuning.

| Src. | Tgt. | K=100 | K=200 | K=500 |
|------|------|-------|-------|-------|
| ant-m | m | **42.2±1.8** | 42.0±0.28 | 41.8±0.7 |
| ant-m-e | m-e | 72.5±2.3 | **74.1±0.50** | 73.8±1.9 |
| ant-m-e | e | 91.9±0.1 | 92.8±2.8 | **93.6±1.3** |
| ant-m-r | m | 40.7±1.2 | 40.8±0.71 | **41.2±0.6** |
| ant-m-r | m-e | 73.0±2.8 | 72.2±3.1 | **74.3±1.6** |
| ant-m-r | e | 90.8±1.6 | 91.3±3.8 | **91.9±0.3** |
| ant-m-e | m | 41.4±0.9 | 41.2±0.6 | **41.5±0.1** |
| ant-m-e | m-e | 71.5±3.1 | 71.1±1.6 | **72.1±5.5** |
| ant-m-e | e | 93.7±0.5 | **94.0±1.1** | 93.9±1.3 |
| Total Score | | 617.7 | 619.5 | **624.1** |

Table G.1: Performance of TCE(OG) on Ant morphology-shift tasks at different $K$ values. Abbrev.: m=medium, m-r=medium-replay, e=expert, m-e=medium-expert. Src./Tgt. denote source/target domain, respectively. Results are mean ± standard deviation over 5 seeds, with the best result in each row shown in **bold**.

## G.2 SOURCE STATE PAIR WITH INVERSE TARGET ACTION

To assess TCE under significant domain gaps, we compare against a baseline named **IQL\*(SwT)** (Source state pairs with Target-inverse actions). In this setting, we train an inverse dynamics model on the target dataset $\mathcal{D}_{\text{tar}}$ and use it to label the actions for source transitions $(s_t, s_{t+1}) \in \mathcal{D}_{\text{src}}$. This creates a synthetic dataset that replaces $\mathcal{D}_{\text{src}}$ for training IQL\*. As shown in Table G.2, IQL\*(SwT) yields comparable performance for medium-quality targets but rapidly degrades as target data becomes more expert due to increased extrapolation error and transition mismatch. TCE, by contrast, consistently outperforms this baseline by expanding state coverage and generating transitions aligned with the target dynamics.

| Src. | Tgt. | OTDF | IQL*(SwT) | TCE(OG) | TCE(NN) |
|------|------|------|-----------|---------|---------|
| half-m | m | 39.1±2.3 | 41.1±0.5 | **44.1±0.2** | 43.8±0.2 |
| half-m | m-e | 35.6±0.7 | 24.6±2.3 | **43.8±0.1** | 43.7±0.1 |
| half-m | e | 10.7±1.2 | 3.3±1.3 | 82.8±0.1 | **85.0±1.2** |
| half-m-r | m | 40.0±1.2 | 38.0±1.5 | **44.0±0.2** | 43.6±0.2 |
| half-m-r | m-e | 34.4±0.7 | 12.9±1.5 | **44.2±0.3** | 43.7±0.1 |
| half-m-r | e | 8.2±2.7 | 2.7±0.1 | **84.4±4** | 77.9±0.2 |
| half-m-e | m | 41.4±0.3 | 41.1±0.2 | **44.2±0.1** | 43.7±0.1 |
| half-m-e | m-e | 35.1±0.6 | 21.0±2.2 | 43.8±0.1 | **43.9±0.5** |
| half-m-e | e | 9.8±.0 | 3.0±2.3 | **85.1±0.8** | 82.6±0.2 |
| hopp-m | m | 11.0±0.9 | 9.9±0.1 | **39.1±0.2** | 8.0±2.3 |
| hopp-m | m-e | 12.6±0.8 | 8.7±0.1 | **29.1±0.1** | 11.0±0.3 |
| hopp-m | e | 10.7±4.7 | 8.2±0.1 | **99.8±0.1** | 10.4±0.1 |
| hopp-m-r | m | 8.7±2.8 | 10.4±1.8 | **49.5±0.1** | 10.7±0.1 |
| hopp-m-r | m-e | 9.7±2.7 | 8.2±1.0 | **17.4±0.3** | 8.3±2.2 |
| hopp-m-r | e | 10.7±2.4 | 8.3±0.5 | **99.7±0.1** | 32.0±6.7 |
| hopp-m-e | m | 7.9±3.2 | 12.0±0.8 | **39.9±0.1** | 14.4±1.9 |
| hopp-m-e | m-e | 9.6±3.5 | 9.2±0.6 | **13.8±0.5** | 8.4±4.8 |
| hopp-m-e | e | 5.9±4.0 | 8.3±0.5 | **99.6±0.1** | 12.7±0.2 |
| **Total Score** | | 341.1 | 270.9 | **1004.3** | 623.8 |

Table G.2: Performance comparison on 18 morphology-shift tasks. Abbrev.: half=HalfCheetah, hopp=Hopper; m=medium, m-r=medium-replay, e=expert, m-e=medium-expert. Src./Tgt. denote source/target domain, respectively. Results are reported as mean ± standard deviation over 5 seeds, with the best result in each row shown in **bold**, second best performance shown in underbar.

# H   PROOF OF THEOREM 1

This section presents the full proof of Theorem 1. We briefly restate the gap bounds between the mixture and target MDPs for completeness.

**Theorem 1** (Restatement). *Let $\eta_{\mathrm{tar}}(\pi)$ and $\eta_{\mathrm{mix}}(\pi)$ denote the expected returns of a policy $\pi$ in the target MDP $\mathcal{M}_{\mathrm{tar}}$ and mixture MDP $\mathcal{M}_{\mathrm{mix}}$, respectively. Suppose*

$$P_{\mathrm{mix}}(\cdot|s,a) = \lambda P_{\mathrm{src}}(\cdot|s,a) + (1-\lambda)\widehat{P}_{\mathrm{tar}}(\cdot|s,a), \qquad \lambda \in [0,1].$$

*For any policy $\pi$, Then the performance gap between the two domains can be bounded from the perspectives of transition dynamics and value functions as follows.*

*Gap bound (transition dynamics).*

$$\eta_{\mathrm{mix}}(\pi) - \eta_{\mathrm{tar}}(\pi) \le \frac{2\gamma r_{\max}}{(1-\gamma)^2} \left( \lambda \, \mathbb{E}_{\rho^\pi_{\mathrm{mix}}} [D_{\mathrm{TV}}(P_{\mathrm{src}}\|P_{\mathrm{tar}})] + (1-\lambda)\mathbb{E}_{\rho^\pi_{\mathrm{mix}}} \left[ D_{\mathrm{TV}}(\widehat{P}_{\mathrm{tar}}\|P_{\mathrm{tar}}) \right] \right), \tag{H.15}$$

*Gap bound (value discrepancy).*

$$\eta_{\mathrm{mix}}(\pi) - \eta_{\mathrm{tar}}(\pi) \le \frac{\gamma}{(1-\gamma)} \Big( \lambda \, \mathbb{E}_{\rho^\pi_{\mathcal{M}_{\mathrm{mix}}}} \left[ \left| \mathbb{E}_{P_{\mathrm{src}}} \left[ V^\pi_{\mathcal{M}_{\mathrm{tar}}}(s') \right] - \mathbb{E}_{P_{\mathrm{tar}}} \left[ V^\pi_{\mathcal{M}_{\mathrm{tar}}}(s') \right] \right| \right]$$
$$+ (1-\lambda) \, \mathbb{E}_{\rho^\pi_{\mathcal{M}_{\mathrm{mix}}}} \left[ \left| \mathbb{E}_{\hat{P}_{\mathrm{tar}}} \left[ V^\pi_{\mathcal{M}_{\mathrm{tar}}}(s') \right] - \mathbb{E}_{P_{\mathrm{tar}}} \left[ V^\pi_{\mathcal{M}_{\mathrm{tar}}}(s') \right] \right| \right] \Big). \tag{H.16}$$

*Proof.* The proofs below utilize the telescoping identity (see Lemma C.1 in Xu et al. (2023)) and are presented in detail for both gap characterizations.

**Gap bound (transition dynamics)** To simplify notation, $P_{\mathrm{mix}}$, $P_{\mathrm{src}}$, and $P_{\mathrm{tar}}$ denote $P_{\mathrm{mix}}(\cdot \mid s,a)$, $P_{\mathrm{src}}(\cdot \mid s,a)$, and $P_{\mathrm{tar}}(\cdot \mid s,a)$ respectively. Similarly, $\rho^\pi_{\mathrm{mix}}$ denotes $\rho^\pi_{\mathrm{mix}}(s,a)$, and $V^\pi_{\mathrm{tar}}$ represents $V^\pi_{\mathrm{tar}}(s')$.

$$\eta_{\mathrm{mix}}(\pi) - \eta_{\mathrm{tar}}(\pi) = \frac{\gamma}{1-\gamma} \mathbb{E}_{\rho^\pi_{\mathrm{mix}}} \left[ \int P_{\mathrm{mix}} V^\pi_{\mathrm{tar}} ds' - \int P_{\mathrm{tar}} V^\pi_{\mathrm{tar}} ds' \right] \qquad \text{(Lemma C.1)}$$

$$= \frac{\gamma}{1-\gamma} \mathbb{E}_{\rho^\pi_{\mathrm{mix}}} \left[ \int (P_{\mathrm{mix}} - P_{\mathrm{tar}}) V^\pi_{\mathrm{tar}} ds' \right]$$

$$\le \frac{\gamma}{1-\gamma} \mathbb{E}_{\rho^\pi_{\mathrm{mix}}} \left[ \int |P_{\mathrm{mix}} - P_{\mathrm{tar}}| \cdot |V^\pi_{\mathrm{tar}}| ds' \right]$$

$$\le \frac{\gamma r_{\max}}{(1-\gamma)^2} \mathbb{E}_{\rho^\pi_{\mathrm{mix}}} \left[ \int |P_{\mathrm{mix}} - P_{\mathrm{tar}}| ds' \right]$$

$$= \frac{2\gamma r_{\max}}{(1-\gamma)^2} \mathbb{E}_{\rho^\pi_{\mathrm{mix}}} [D_{\mathrm{TV}}(P_{\mathrm{mix}}\|P_{\mathrm{tar}})]$$

$$= \frac{2\gamma r_{\max}}{(1-\gamma)^2} \mathbb{E}_{\rho^\pi_{\mathrm{mix}}} \left[ D_{\mathrm{TV}}((\lambda P_{\mathrm{src}} + (1-\lambda)\hat{P}_{\mathrm{tar}})\|P_{\mathrm{tar}}) \right]$$

$$\le \frac{2\gamma r_{\max}}{(1-\gamma)^2} \mathbb{E}_{\rho^\pi_{\mathrm{mix}}} \left[ D_{\mathrm{TV}}(\lambda P_{\mathrm{src}}\|P_{\mathrm{tar}}) + D_{\mathrm{TV}}((1-\lambda)\hat{P}_{\mathrm{tar}}\|P_{\mathrm{tar}}) \right]$$

$$= \frac{2\gamma r_{\max}}{(1-\gamma)^2} \left( \lambda \mathbb{E}_{\rho^\pi_{\mathrm{mix}}} [D_{\mathrm{TV}}(P_{\mathrm{src}}\|P_{\mathrm{tar}})] + (1-\lambda)\mathbb{E}_{\rho^\pi_{\mathrm{mix}}} \left[ D_{\mathrm{TV}}(\hat{P}_{\mathrm{tar}}\|P_{\mathrm{tar}}) \right] \right)$$

**Gap bound(value discrepancy)**

$$\eta_{\mathrm{mix}}(\pi) - \eta_{\mathrm{tar}}(\pi) = \frac{\gamma}{1-\gamma}\mathbb{E}_{\rho^{\pi}_{\mathrm{mix}}(s,a)}\left[\int_{s'} P_{\mathrm{mix}}(s'|s,a)V^{\pi}_{\mathcal{M}_{\mathrm{tar}}}(s')ds'\right.$$
$$\left.-\int_{s'} P_{\mathrm{tar}}(s'|s,a)V^{\pi}_{\mathcal{M}_{\mathrm{tar}}}(s')ds'\right] \qquad \text{(Lemma C.1)}$$

$$= \frac{\gamma}{1-\gamma}\mathbb{E}_{\rho^{\pi}_{\mathcal{M}_{\mathrm{mix}}}}\left[\mathbb{E}_{P_{\mathrm{mix}}}[V^{\pi}_{\mathcal{M}_{\mathrm{tar}}}(s')] - \mathbb{E}_{P_{\mathrm{tar}}}[V^{\pi}_{\mathcal{M}_{\mathrm{tar}}}(s')]\right]$$

$$= \frac{\gamma}{1-\gamma}\mathbb{E}_{\rho^{\pi}_{\mathcal{M}_{\mathrm{mix}}}}\left[\mathbb{E}_{P_{\mathrm{mix}}}[V^{\pi}_{\mathcal{M}_{\mathrm{tar}}}(s')] - \mathbb{E}_{P_{\mathrm{tar}}}[V^{\pi}_{\mathcal{M}_{\mathrm{tar}}}(s')]\right]$$

$$\leq \frac{\gamma}{1-\gamma}\mathbb{E}_{\rho^{\pi}_{\mathcal{M}_{\mathrm{mix}}}}\left[\left|\mathbb{E}_{P_{\mathrm{mix}}}[V^{\pi}_{\mathcal{M}_{\mathrm{tar}}}(s')] - \mathbb{E}_{P_{\mathrm{tar}}}[V^{\pi}_{\mathcal{M}_{\mathrm{tar}}}(s')]\right|\right]$$

$$= \frac{\gamma}{1-\gamma}\mathbb{E}_{\rho^{\pi}_{\mathcal{M}_{\mathrm{mix}}}}\left[\left|\lambda\mathbb{E}_{P_{\mathrm{src}}}[V^{\pi}_{\mathcal{M}_{\mathrm{tar}}}(s')]\right.\right.$$
$$\left.\left.+ (1-\lambda)\mathbb{E}_{\hat{P}_{\mathrm{tar}}}[V^{\pi}_{\mathcal{M}_{\mathrm{tar}}}(s')] - \mathbb{E}_{P_{\mathrm{tar}}}[V^{\pi}_{\mathcal{M}_{\mathrm{tar}}}(s')]\right|\right]$$

$$= \frac{\gamma}{1-\gamma}\mathbb{E}_{\rho^{\pi}_{\mathcal{M}_{\mathrm{mix}}}}\left[\left|\lambda(\mathbb{E}_{P_{\mathrm{src}}}[V^{\pi}_{\mathcal{M}_{\mathrm{tar}}}(s')] - \mathbb{E}_{P_{\mathrm{tar}}}[V^{\pi}_{\mathcal{M}_{\mathrm{tar}}}(s')])\right.\right.$$
$$\left.\left.+ (1-\lambda)(\mathbb{E}_{\hat{P}_{\mathrm{tar}}}[V^{\pi}_{\mathcal{M}_{\mathrm{tar}}}(s')] - \mathbb{E}_{P_{\mathrm{tar}}}[V^{\pi}_{\mathcal{M}_{\mathrm{tar}}}(s')])\right|\right]$$

$$\leq \frac{\gamma}{1-\gamma}\mathbb{E}_{\rho^{\pi}_{\mathcal{M}_{\mathrm{mix}}}}\left[\lambda\left|\mathbb{E}_{P_{\mathrm{src}}}[V^{\pi}_{\mathcal{M}_{\mathrm{tar}}}(s')] - \mathbb{E}_{P_{\mathrm{tar}}}[V^{\pi}_{\mathcal{M}_{\mathrm{tar}}}(s')]\right|\right.$$
$$\left.+ (1-\lambda)\left|\mathbb{E}_{\hat{P}_{\mathrm{tar}}}[V^{\pi}_{\mathcal{M}_{\mathrm{tar}}}(s')] - \mathbb{E}_{P_{\mathrm{tar}}}[V^{\pi}_{\mathcal{M}_{\mathrm{tar}}}(s')]\right|\right]$$

$$= \frac{\gamma}{1-\gamma}\left(\lambda\mathbb{E}_{\rho^{\pi}_{\mathcal{M}_{\mathrm{mix}}}}\left[\left|\mathbb{E}_{P_{\mathrm{src}}}[V^{\pi}_{\mathcal{M}_{\mathrm{tar}}}(s')] - \mathbb{E}_{P_{\mathrm{tar}}}[V^{\pi}_{\mathcal{M}_{\mathrm{tar}}}(s')]\right|\right]\right.$$
$$\left.+ (1-\lambda)\mathbb{E}_{\rho^{\pi}_{\mathcal{M}_{\mathrm{mix}}}}\left[\left|\mathbb{E}_{\hat{P}_{\mathrm{tar}}}[V^{\pi}_{\mathcal{M}_{\mathrm{tar}}}(s')] - \mathbb{E}_{P_{\mathrm{tar}}}[V^{\pi}_{\mathcal{M}_{\mathrm{tar}}}(s')]\right|\right]\right)$$

$$\square$$

The derived gap bound highlights two distinct avenues for reducing the performance discrepancy. The term involving $D_{\mathrm{TV}}(P_{\mathrm{src}}||P_{\mathrm{tar}})$ suggests that the gap is tightened when source transitions align well with the target dynamics, while the term involving $D_{\mathrm{TV}}(\hat{P}\mathrm{tar}||P\mathrm{tar})$ indicates that minimizing the generative error is crucial. Moreover, the mixture coefficient $\lambda$ controls the trade-off between these two terms, enabling us to attenuate the influence of mismatched source dynamics while leveraging accurate target-like generations to tighten the overall gap bound.

