# OpenReview forum: "Two-Stage Coverage Expansion for Cross-Domain Offline Reinforcement Learning via Score-Based Generative Modeling"
_ICLR.cc/2026/Conference — Submitted to ICLR 2026_

### Official Review · Reviewer_z9Ke · 2025-10-31

**Soundness:** 2
**Presentation:** 3
**Contribution:** 2
**Rating:** 2
**Confidence:** 4

**Summary:**

This paper presents a data augmentation method to solve the cross-domain adaptation RL problem. Specifically, the two-stage method comprises a state expansion module and a transition module, which generate new target-like data. Further, they provide a data filtering method to filter the generated data.

**Strengths:**

The paper is well written and easy to follow. The motivation is clear to me, and the solution seems novel and fits the problem. Figure 1 and the visualization appear clear to me, which effectively states the problem.  The experimental results appear promising under specific conditions.

**Weaknesses:**

1. The experimental setting is not well stated (maybe I am missing something, or at least you should state that in the table caption?). What is the shift level the paper are using in the experiment section? The ODRL benchmark has easy/medium/hard for kinematics and morphology shift and 0.1/0.5/2.0/5.0 for gravity? For gravity shift, the paper seems to report only a 0.5 shift level, and friction shift is not included (I found it very far away from the table in the XML code). I suggest that more shift levels should be preferred, and a large shift case should be included, as the author claims that they can improve in a large shift case such as gravity/friction 5.0/0.5 or morphology/kinematics shift.

2. Though the motivation of the paper is clear to me, I am not very clear about the motivation of the method itself. For eaxmple, why do you need to expand the state instead of only using the source/target state to do the second stage? Regarding the transition, why do you only learn the (s_t, s_{t+1}) instead of including the action, (though it might be hard as I can imagine)? If you ignore the action here, it seems that you are only generating the some "state pair", regardless the action.

3. Figure 3 (a) looks good to me, and it is good to have such a visualization. It seems that the generated state avoids all the source states, showing that it learns something. However, I wonder whether the generated states are actually the target states or just something in between the source and target states? Also, is it possible that you have some figure like Fig. (a) to show what the generated transition looks like, whether it is really some new transition instead of only mimicking/overfitting the target data?

4. Another question is regarding the method itself.  As stated in (2) you ignore the action here, it seems that you are only generating the some "state pair", regardless the action. And it seems to me that you are given some state pair and do the inverse state to get the action. Then, why do you still need such complicated generation process, instead of directly inverse the source state pair (or some trivial solution to get some state pair) to get the target action and filter out data. It seems to me that the inverse action part is something that really make this method works.  Because given any state pair, you can always use the inverse action to make the transition look like target data and then filter out those bad data, it is just a matter of computational efficiency. So, I suggest that more experiments are needed.

**Questions:**

See weakness.

---

> ### Author Response · Authors · 2025-11-22
>
> **Weakness 1 (Environmental setup):**
>
> We thank the reviewer for the helpful suggestion. To clarify our experimental setup, we adopted exactly the same environment configuration as OTDF. We have made this design choice explicit in Appendix C.2 and further refined the discussion by providing a more detailed quantitative description of how the domains differ within the benchmark.
>
> In addition, as suggested by the reviewer, we extended the analysis in Appendix D.2 to include more extreme domain shift scenarios in the ODRL benchmark, such as the gravity x5. In Table D.2, we report a comparison among IQL*, OTDF, and TCE under these severe domain differences, and we observe that TCE consistently outperforms OTDF even in such challenging settings. We believe these results highlight the practical advantage of TCE, and we appreciate the reviewer’s encouragement to strengthen this aspect.
>
> **Weakness 2 (Motivation of TCE):**
>
> We thank the reviewer for the thoughtful question. In our framework, actions are not ignored; rather, TCE first generates possible next states conditioned on a given TCE state, and then an inverse model is used to recover the corresponding actions. Concretely, if the full data distribution is $P(s_t, a_t, s_{t+1})$, one could follow the factorization $P(s_{t+1} \mid s_t, a_t) P(s_t, a_t)$ and directly model next states from $(s_t, a_t)$, as the reviewer suggested. **However, it is equally valid to adopt the alternative factorization $P(s_t, s_{t+1}) P(a_t \mid s_t, s_{t+1})$, in which we first generate state pairs $(s_t, s_{t+1})$ and then infer $a_t$ via an inverse model.** Our method follows this latter viewpoint. We designed our approach to encourage generalization by learning a generative model over state pairs with shared structure, while using the inverse model as a standard mechanism for action recovery. This implementation aligns with the intended probabilistic formulation, and we hope this clarifies our design choice.
>
> **Weakness 3 (Reliability of two-stage approach):**
>
> We appreciate the reviewer’s careful observation that our method relies on the reliability of generated samples. To assess the reliability of our two stage approach, we first evaluate TCE transition generation by measuring the distributional discrepancy between generated and true target transitions as a function of $y_{\max}$, where $y_{\max}$ serves as a coverage control parameter, as illustrated in Fig. 3(a). From these results, we observe in Fig. 3(b) that when $y_{\max} \le 0.2$ the KL divergence remains small, indicating that the generated transitions are reasonably faithful to the target dynamics in this regime.
>
> Furthermore, to examine how reliable the states generated from mixture states are for different values of $y_{\max}$, we additionally compute their errors with respect to both real target and source states and report the results in Fig. 3(b). We find that for $0 < y_{\max} < 1$ the error is comparable to that obtained when using pure target ($0$) or pure source ($1$) labels, which supports that mixture state generation based on $y_{\max}$ is sufficiently reliable. In light of this analysis, the mixture model is better understood not as producing “intermediate” states between the source and target, but as generating a diverse set of states that are each consistent with either the source or the target distribution, while the target transition model focuses on generating plausible next states that approximate the true target transition probability.
>
> **(Reliability of reward/inverse model):**
>
> In addition, we agree that further consideration is needed for the reward and inverse models. For the inverse model, a direct comparison is difficult because there is no explicit target policy that maps states to actions. For the reward model, however, we explicitly compare the rewards predicted for generated samples with those from the true reward function, as reported in Fig. B.3. This reward analysis confirms that up to $y_{\max} = 0.2$ the errors with respect to real samples remain small, which supports the conclusion that the generated samples used by TCE are sufficiently reliable in the regime in which we apply the method.
>
> **Weakness 4 (Additional simple baseline):**
>
> We thank the reviewer for the insightful question. As requested, we performed an ablation that augments source state pairs with inverse target actions under morphology shifts (Appendix G.2). This baseline remains competitive at medium target quality but its performance drops notably as the target dataset approaches expert level, suggesting that large domain gaps amplify OOD extrapolation errors in the inverse model. **These results indicate that inverse actions alone are insufficient to bridge large domain gaps. By contrast, TCE first establishes sufficient target coverage and then generates target-aligned transitions, mitigating both coverage and transition errors.** We hope this clarifies the necessity of TCE beyond the inverse-action baseline.

---

### Official Review · Reviewer_hkUn · 2025-11-01

**Soundness:** 1
**Presentation:** 3
**Contribution:** 4
**Rating:** 4
**Confidence:** 3

**Summary:**

The paper propose a novel two-stage network approach for cross-domain RL. In stage 1, train a mixture SGM to expand state-space coverage by leveraging rich source data and in stage 2 uses a conditional SGM to align the generated transitions with the target-domain dynamics. Followed by auxiliary model to create the corresponding action and reward, the paper are able to curate an augmented dataset.

**Strengths:**

1. Novel and principled methodological design: the two-stage design is a novel and elegant solution to the data transfer problem in cross-domain RL. By separating the learning of state coverage from the learning of target dynamics, the work offers a clean, architectural approach for data augmentation in RL that moves beyond simple distance-metric-based source-data selection.

2. Strong empirical performance: The experiments result strongly suggests that the method achieves "substantial gains" and "consistently outperforms state-of-the-art cross-domain RL baselines" even under large domain gaps and extremely small target datasets. This empirical strength on a critical, real-world problem warrants serious consideration.

**Weaknesses:**

1. Lack of rigorous mathematical proof: The paper does not provide a formal boundedness analysis on the resulting distribution of the augmented data which is mandatory for any method based on synthesis in offline RL. It does not introduce any theoretical modification to the policy constraint or pessimism. The policy still potentially risks catastrophic Extrapolation Error when acting greedily on the augmented data

2. No guarantees of filtering: The reliance on "Z-score-based filtering" can not guarantee that the distance like KL divergence between the augmented dataset and the true target dynamics are provably bounded. The filtering itself could inadvertently prune high-value, but naturally low-likelihood, target transitions, thereby introducing a new bias.

3. Lake of enough sensitivity analysis: Score-based models are intrinsically sensitive to discretization steps, noise schedules, and weighting functions. By chaining two score-matching procedures, the overall framework's sensitivity is compounded, making the method a hyperparameter tuning nightmare. The lack of comprehensive sensitivity analysis severely limits the reproducibility and trust in the empirical results.

4. Potential computional burden: Training two score networks + the auxiliary models adds significant computional burden.

**Questions:**

1. Provide a theoretical result that demonstrates that the combination of the two-stage SGM and the Z-score filtering step provides a bound on the KL divergence between $D_{aug}, D_{tar}$ or similar distance. What is the theoretical justification for the Z-score threshold choice?

2. Present an ablation study comparing the full TCE against a simpler, single conditional SGM, $p(s, s' | a)$, that models the entire transition directly, or a non-mixture-based state SGM. Is the significant computational overhead strictly necessary for the claimed performance gains?

3. Provide a comprehensive sensitivity analysis for the SGM's key hyperparameters like the noise schedule $\sigma(\tau)$ and the number of discretization steps $K$. How robust is the final performance to these choices?

---

> ### Author Response · Authors · 2025-11-22
>
> **Weakness 1 (Theoretical foundation of TCE):**
>
> We sincerely appreciate the reviewer’s insightful comments on the theoretical foundation of TCE and fully share the reviewer’s concerns. In response, during the rebuttal period we have strengthened the theoretical justification and incorporated a more in-depth analysis in the revised version. In particular, **Theorem 1 provides a gap bound analysis that highlights the necessity of TCE by showing that a mixture of source data and TCE generated samples can achieve a tighter performance gap bound than using only source data.** Motivated by this result, we additionally compare (i) TCE (OG), which uses only generated samples, and (ii) TCE (NN), which combines a distance based source selection, in the spirit of prior work, with TCE generated samples. These ablations, together with a discussion on the reliability of generated samples, offer a more in depth understanding of how TCE improves performance and clarify its distinct role beyond traditional distance based methods. We believe these additions substantially enhance both the quality and the novelty of the paper, and we are grateful to the reviewer for encouraging us to make these points more explicit. We will address the reliability and choice of $y_{\max}$ in our response to the question below.
>
> **Weakness 2/Question 1 (Z-score filtering):**
>
> With respect to Z-score filtering, we thank the reviewer for raising this point. We follow prior generative approaches [R.1] that also use this technique to detect outliers.
> **In the revised manuscript, we provide a comprehensive reliability analysis in Figure 3(b) and Appendix F, quantitatively demonstrating that the samples rejected by Z-score filtering consistently exhibit significantly higher transition and reward prediction errors.** While we did not perform this analysis for every single environment, we observed a consistent trend across most representative tasks, as detailed in Appendix F. This finding confirms that Z-score filtering does not merely remove statistical outliers but effectively identifies and discards low-reliability samples that contradict the learnable target dynamics.
>
> We would also like to clarify the role of Z-score filtering in our method. Z-score filtering is used as a practical algorithmic tool to remove abnormal outliers that deviate substantially from the main state distribution, rather than as a component of our theoretical analysis. The theoretical properties of TCE are established in Theorem 1, as discussed above, where we assume that the model approximation errors are relatively small compared with the discrepancy between the two domains, and the reliability analysis provides a concrete empirical assessment of these approximation errors in practice. We hope this clarification is helpful to the reviewer.
>
> **Weakness 3/Question 3 (Hyperparameter sensitivity):**
>
> We appreciate the reviewer's thoughtful comment regarding hyperparameter search. For the SGM parameters, we focus our analysis on the most influential hyperparameter, namely the noise schedule. In particular, for varying denoising steps $K \in \{100, 200, 500\}$, we have reported the performance results in Appendix G.1. As shown in Table G.1, while increasing $K$ yields modest performance improvements, the differences remain minor across all settings, indicating that our method achieves consistently strong performance and is not highly sensitive to this hyperparameter.
>
> **Weakness 4 (Computational cost):**
>
> We thank the reviewer for this valuable comment. As shown in the computational cost analysis in Appendix E, training the SGM model indeed introduces additional computational cost, and we agree that this constitutes a nontrivial overhead. At the same time, Theorem 1 clarifies why TCE is necessary in our setting, and empirically our method achieves substantially higher performance than existing alternatives, which, to our knowledge, cannot reach a comparable level under the same cross domain offline setup. We therefore regard this additional cost as a necessary trade off for the performance gains provided by TCE, and we hope this explanation helps contextualize the computational overhead.
>
> **Question 2 (Model structure):**
>
> We thank the reviewer for this helpful suggestion. As described in Eq. 9, the two models in our framework play fundamentally different roles and are trained on different data, so they cannot be merged into a single generator. In particular, the mixture model uses both source and target data to model a mixture of the two distributions, whereas the transition model is trained only on target transitions in order to accurately capture the target dynamics. Because their objectives and training distributions are inherently different, it is important to keep these two networks separate.
>
> [R.1] Shin, et al. "Strainer GAN: Filtering out Impurity Samples in GAN Training." SIGGRAPH Asia 2024 Posters. 2024. 1-2.

---

### Official Review · Reviewer_RPuV · 2025-11-03

**Soundness:** 3
**Presentation:** 2
**Contribution:** 2
**Rating:** 6
**Confidence:** 3

**Summary:**

This paper proposes TCE to resolve the distribution mismatch in resuing source transitions. TCE expands state coverage first and aligns transitions with the target domain. Experiments show that TCE outperforms cross-domain baselines on several benchmarks.

**Strengths:**

- The paper introduces a clear and well-motivated two-stage generative design, where state and transition modeling are decoupled, leveraging conditional score-based diffusion modeling to manage both state support expansion and careful transition dynamics alignment.

- The results section offers extensive benchmarking across 36 cross-domain morphology, kinematic, and gravity shift tasks in MuJoCo, comparing TCE to numerous strong baselines (IQL*, DARA, BOSA, SRPO, IGDF, OTDF). TCE solidly outperforms these across most scenarios, especially where the target data is scarce or the domain gap is high.

**Weaknesses:**

- The quality of synthesized transitions is at the mercy of the inverse dynamics and reward prediction models, both trained only on small target datasets. Given the known overfitting risks and distributional shifts, further justification or diagnostic evaluation of the labeling process (as the final step before RL training) would be helpful. There is, for instance, no explicit quantitative evaluation of the action/reward labeling accuracy on held-out real transitions.

- The selection of baselines is solid for cross-domain settings. However, the comparison omits a few important general offline RL methods (e.g., MOReL, which also emphasizes pessimism and support constraints) that could provide a useful baseline reference, helping to ground the observed performance gains.

- Several directly relevant recent works are omitted from the related works. Notably, recent papers on nearest-neighbor-guided diffusion, reverse dynamics-based cross-domain offline RL, and hybrid robust RL are not discussed or empirically compared. This limits the ability of the reader to precisely situate TCE’s advances and also misses opportunities to critique or learn from closely related design choices.

**Questions:**

1. Can the authors provide more systematic evidence (e.g., statistical plots or empirical tabulations) for the robustness of TCE to $y_{\max}$ and $z_{\mathrm{th}}$ across all domains/agents and more than just a handful? Are there settings where improper tuning of these parameters substantially reduces policy performance?
2. Can the authors demonstrate qualitative or even hand-checked inspection of generated transitions for high-dimensional states (esp. Ant), to substantiate that coverage expansion does not admit unrealistic/unfeasible samples (beyond Z-score filtering)?
3. How scalable is the approach in both data generation (time/memory) and RL training if (a) state/action spaces are significantly higher-dimensional, or (b) there are “many” source domains? What are the concrete bottlenecks and trade-offs?
4. Could the authors explicitly discuss and, if possible, empirically compare to the directly relevant recent diffusion-based, reverse dynamics, and hybrid robust cross-domain RL works listed above?
5. What's the performance of the proposed method without the KL regularization term? Is this term added to other baselines?

---

> ### Author Response · Authors · 2025-11-22
>
> **Weakness 1/Question 1 (Reliability of generated samples):**
>
> We appreciate the reviewer’s careful observation that our method relies on the reliability of generated samples. To assess the reliability of our two stage approach, we first evaluate TCE transition generation by measuring the distributional discrepancy between generated and true target transitions as a function of $y_{\max}$, where $y_{\max}$ serves as a coverage control parameter, as illustrated in Fig. 3(a). **From these results, we observe in Fig. 3(b) that when $y_{\max} \le 0.2$ the KL divergence remains small, indicating that the generated transitions are reasonably faithful to the target dynamics in this regime.**
>
> In addition, we agree that further consideration is needed for the reward and inverse models. For the inverse model, a direct comparison is difficult because there is no explicit target policy that maps states to actions. For the reward model, however, we explicitly compare the rewards predicted for generated samples with those from the true reward function, as reported in Fig. B.3. This reward analysis confirms that up to $y_{\max} = 0.2$ the errors with respect to real samples remain small, which supports the conclusion that the generated samples used by TCE are sufficiently reliable in the regime in which we apply the method.
>
> **(Z-score filtering):**
>
> With respect to Z-score filtering, we thank the reviewer for raising this point. We follow prior generative approaches [R.1] that also use this technique to detect outliers.
> **In the revised manuscript, we provide a comprehensive reliability analysis in Figure 3(b) and Appendix F, quantitatively demonstrating that the samples rejected by Z-score filtering consistently exhibit significantly higher transition and reward prediction errors.** While we did not perform this analysis for every single environment, we observed a consistent trend across most representative tasks, as detailed in Appendix F. This finding confirms that Z-score filtering does not merely remove statistical outliers but effectively identifies and discards low-reliability samples that contradict the learnable target dynamics. While this procedure does not guarantee that all remaining samples are perfectly matched to the true state distribution, it at least discards samples that are likely to be unreliable. Consistent with this analysis, the component evaluation in Table 3 empirically suggests that the proposed Z-score filtering is beneficial for training, and we hope this clarification helps address the reviewer’s concern.
>
> **Weakness 2/3, Question 4 (Additional baselines):**
>
> Regarding the choice of baselines, as can be seen in the paper, we already compare against a wide range of methods specifically designed for the cross domain setup. In contrast, MOReL is formulated for the standard offline RL setting under a single dynamics model and does not address cross domain transfer at all, so it is not an appropriate baseline for our problem. For the other cross domain methods the reviewer mentioned, since the exact references were not specified, we respond based on representative examples we are aware of. Hybrid Robust RL (HYDRO, [R.2]) assumes access to an online source task, whereas our setting uses only offline source data, which makes our setup strictly more challenging and also prevents a direct comparison under the same assumptions. Nearest Neighbor Diffusion (DmC, [R.3]) was published at ECAI 2025, so it was not available at the time of our submission, and to our knowledge there is currently no public implementation, we conducted a comparative analysis based on the kinematic shift results reported in the DmC paper. As detailed in Appendix D.3, TCE achieves highly competitive performance, demonstrating superior or comparable results in most tasks. This comparison confirms that our method remains robust and effective even against the latest advancements.

---

> ### Author Response · Authors · 2025-11-22
>
> **Question 2 (sample fidelity):**
>
> \tcr{We appreciate the reviewer’s question. In the revised manuscript, Figure 3(b) and Appendix F provide a quantitative reliability analysis of Z-score filtering by comparing transition and reward prediction errors between retained samples and those rejected as outliers across representative environments. These results show that Z-score filtering consistently removes samples with substantially higher errors, indicating that it preferentially discards low-reliability transitions. While this analysis does not guarantee perfectly optimal filtering in all cases, it supports that Z-score filtering effectively excludes unreliable samples and thereby improves the overall quality of the augmented dataset. We hope these checks help address the reviewer’s concern.}
>
> **Question 3 (computational cost):**
>
> We thank the reviewer for the insightful comment. We analyzed computational costs on HalfCheetah, Hopper, Walker2d, and Ant environments with 2GB GPU memory each. Training took about 1h55m and sampling 10m for HalfCheetah, Hopper, and Walker2d; for Ant, whose state dimension is nearly six times larger than the others, training took 2h5m and sampling 12m. Details are provided in Appendix E. These results demonstrate reasonable and scalable computational overhead. We hope this clarifies any concerns about computational overhead.
>
> **Question 5 (Ablation for KLD):**
>
> To address the reviewer’s question, we additionally report in Table 3 the performance of our method in the component evaluation setting without the policy regularizer. Removing the KL term does lead to a certain degradation, indicating that policy regularization has some effect, but as shown in the new results, the overall performance drop is modest. This suggests that our method is not overly dependent on the policy regularizer, since the target-style transition data generated by TCE already cover a wide range of stages of the task, which stabilizes learning even without this regularization. We hope this clarification helps address the reviewer’s concern.
>
> [R.1] Shin et al. "Strainer GAN: Filtering out Impurity Samples in GAN Training." SIGGRAPH Asia 2024 Posters. 2024. 1-2.
>
> [R.2] Van et al. "Hybrid Cross-Domain Robust Reinforcement Learning." ECML Cham: Springer Nature Switzerland, 2025.
>
> [R.3] Van et al. "DmC: Nearest Neighbor Guidance Diffusion Model for Offline Cross-domain Reinforcement Learning." arXiv preprint arXiv:2507.20499 (2025).

---

### Official Review · Reviewer_fFKD · 2025-11-03

**Soundness:** 3
**Presentation:** 3
**Contribution:** 2
**Rating:** 4
**Confidence:** 4

**Summary:**

In this paper, the authors study cross-domain offline reinforcement learning where the source and target MDPs share \((\mathcal S, \mathcal A, R, \gamma)\) but differ in transition dynamics \(P_{src} \neq P_{tar}\), and the target dataset is small. They propose Two-stage Coverage Expansion, a score-based generative framework with SDEs: a mixture-based state score network first expands target state coverage via controllable source–target interpolation, and a target-transition score network then generates next states aligned with target dynamics. After generation, Z-score filtering removes outliers, inverse dynamics and reward models reconstruct actions and rewards, and the resulting synthetic dataset augments for offline policy learning with IQL plus KL regularization to stay close to target behavior. Experiments on MuJoCo tasks under morphology, kinematic, and gravity shifts show that naive source data reuse can harm performance when domain gaps are large, while TCE improves returns by balancing coverage and dynamics consistency. Overall, TCE provides a  pipeline that decouples coverage expansion from transition alignment to enhance data efficiency in cross-domain offline RL.

**Strengths:**

1. The experimental evaluation is extensive and well-executed: the authors test TCE across multiple MuJoCo domains , and provide detailed ablations on key hyperparameters.
2.    The theoretical part is well-grounded, the paper carefully defines builds the method on established SDE-based score generation theory, and shows step by step how the two score networks work together to expand coverage and match target dynamics.

**Weaknesses:**

1.   The TCE framework relies on two separately trained score-based generative models, which increases implementation complexity and training cost. The coupling between these two stages is not formally analyzed, and potential error propagation from Stage 1 to Stage 2 is not well quantified. Moreover, the subsequent construction of \(D_{gen}\) through simple inverse dynamics and reward prediction appears somewhat ad hoc, lacking a more principled treatment of uncertainty or consistency. Overall, while the data generation pipeline is elaborate, the offline RL component itself remains relatively standard, making the overall architecture feel unbalanced—most design novelty lies in data synthesis rather than policy learning.
2.   The mixture-based state sampling in Stage 1 depends heavily on the hyperparameter \(y_{\max}\), yet no principled way to set or adapt this parameter is provided. The balance between “coverage expansion” and “dynamics consistency” is treated empirically rather than theoretically.

3.   The coverage expansion in Stage-1 focuses on enlarging the statistical support of the target state distribution, but it does not guarantee physical reachability. Even if the generated \(\hat{s}_t\) lies within a statistically plausible region after Z-score filtering, it may still be dynamically infeasible under the target environment’s transition dynamics. Since Z-score is purely a statistical criterion, it cannot ensure that generated states belong to the reachable set of the underlying MDP. A more principled approach would explicitly incorporate reachability constraints or learned feasibility critics to maintain physical consistency.

**Questions:**

Q1: The paper defines the cross-domain setting as sharing the same state, action, and reward spaces while differing only in transition dynamics. Although this assumption can be satisfied in MuJoCo-style simulations, real-world cross-domain scenarios often involve variations in observation or action spaces. Would this simplification make the problem definition too narrow or less representative of practical settings?

Q2: If there exist slight mismatches in state spaces (e.g., limited joint angles or missing body parts in Ant variants), can the proposed TCE framework still function effectively, given that it assumes perfectly aligned state–action representations across domains?

Q3: The baselines do not include decision-transformer-based offline RL methods such as Meta Decision Transformer or Prompt Decision Transformer, which explicitly handle domain variations in both reward functions and dynamics. Could the authors clarify why such methods were not compared, and whether TCE could be integrated with or outperform them under the same problem setting?

Q4: In Section 4.3, the inverse dynamics and reward models are trained using only \((s, s')\) pairs. What is the rationale behind this design choice, and is the assumption that the reward can be accurately inferred from \((s, s')\) alone theoretically sound in tasks where

---

> ### Author Response · Authors · 2025-11-22
>
> **Weakness 1/2 (Theoretical foundation of TCE):**
>
> We sincerely appreciate the reviewer’s insightful comments on the theoretical foundation of TCE and fully share the reviewer’s concerns. In response, during the rebuttal period we have strengthened the theoretical justification and incorporated a more in-depth analysis in the revised version. In particular, Theorem 1 provides a gap bound analysis that highlights the necessity of TCE by showing that a mixture of source data and TCE generated samples can achieve a tighter performance gap bound than using only source data. Motivated by this result, we additionally compare (i) TCE (OG), which uses only generated samples, and (ii) TCE (NN), which combines a distance based source selection, in the spirit of prior work, with TCE generated samples. These ablations, together with a discussion on the reliability of generated samples, offer a more in depth understanding of how TCE improves performance and clarify its distinct role beyond traditional distance based methods. We believe these additions substantially enhance both the quality and the novelty of the paper, and we are grateful to the reviewer for encouraging us to make these points more explicit. We will address the reliability and choice of $y_{\max}$ in our response to the question below.
>
> **(Reliability of two stage approach):**
>
> We appreciate the reviewer’s careful observation that our method relies on the reliability of generated samples, which is consistent with the assumption in Theorem 1 that the approximate dynamics $\hat P_{\mathrm{tar}}$ should be close to the true dynamics $P_{\mathrm{tar}}$. To assess the reliability of our two stage approach, we first evaluate TCE transition generation by measuring the distributional discrepancy between generated and true target transitions as a function of $y_{\max}$, where $y_{\max}$ serves as a coverage control parameter, as illustrated in Fig. 3(a). **From these results, we observe in Fig. 3(b) that when $y_{\max} \le 0.2$ the KL divergence remains small, indicating that the generated transitions are reasonably faithful to the target dynamics in this regime.**
>
> Furthermore, to examine how reliable the states generated from mixture states are for different values of $y_{\max}$, we additionally compute their errors with respect to both real target and source states and report the results in Fig. 3(b). We find that for $0 < y_{\max} < 1$ the error is comparable to that obtained when using pure target ($0$) or pure source ($1$) labels, which supports that mixture state generation based on $y_{\max}$ is sufficiently reliable. In light of this analysis, the mixture model is better understood not as producing “intermediate” states between the source and target, but as generating a diverse set of states that are each consistent with either the source or the target distribution, while the target transition model focuses on generating plausible next states that approximate the true target transition probability.
>
>
> **(Reliability of reward/inverse model):**
>
> In addition, we agree that further consideration is needed for the reward and inverse models. For the inverse model, a direct comparison is difficult because there is no explicit target policy that maps states to actions. For the reward model, however, we explicitly compare the rewards predicted for generated samples with those from the true reward function, as reported in Fig. B.3. This reward analysis confirms that up to $y_{\max} = 0.2$ the errors with respect to real samples remain small, which supports the conclusion that the generated samples used by TCE are sufficiently reliable in the regime in which we apply the method.

---

> ### Author Response · Authors · 2025-11-22
>
> **Weakness 3 (Stage 1/Z-score filtering):**
>
> We thank the reviewer for raising this point. Regarding the reviewer’s concern about the coverage extension of Stage 1, as discussed above we empirically verified that the generated mixture states remain sufficiently close to true target and source states, which alleviates the concern about their reachability.
> With respect to Z-score filtering, we refer prior generative approaches[R.1] that also use this technique to detect outliers. **In the revised manuscript, we provide a comprehensive reliability analysis in Fig. 3(b). and Appendix G, quantitatively demonstrating that the samples rejected by Z-score filtering consistently exhibit significantly higher transition and reward prediction errors compared to the retained samples.** This finding confirms that Z-score filtering does not merely remove statistical outliers but effectively identifies low-reliability samples that deviate from the learnable target dynamics.
> While this procedure does not guarantee that all remaining samples are perfectly matched to the true state distribution, it successfully discards samples that are likely to be unreliable and harmful to policy learning. Consistent with this analysis, the component evaluation in Table 3 empirically suggests that the proposed Z-score filtering is beneficial for training, and we hope this clarification helps address the reviewer’s concern.
>
>
> **Question 1/2 (Assumptions on environment changes):**
>
> We thank the reviewer for raising this important point about the assumptions on environment changes. As the reviewer correctly noted, our work primarily focuses on the standard cross-domain offline setup where the main difference between domains lies in the transition dynamics, while the state and action spaces remain compatible. If, instead, the two environments were entirely different so that the state, observation, or action spaces are not aligned at all, for example an Ant with a broken leg or a transfer from a cheetah environment to an ant environment, then additional representation learning for distribution matching would indeed be necessary, as considered in cross-domain imitation learning methods such as [R.2,R.3]. A thorough treatment of such heterogeneous state or action spaces is therefore beyond the main scope of this paper, which concentrates on dynamics changes under shared spaces, but we agree that this is a very interesting and significantly more challenging direction for future work.
>
> **Question 3 (Comparison with DT-based methods):**
>
> Regarding approaches such as Meta Decision Transformer, these methods are designed to leverage offline datasets that already contain variations in reward or dynamics in order to generalize to a wide range of downstream test tasks, rather than to address the cross-domain RL setting with a single source domain and a single target domain. In this sense, their problem formulation and offline data setup are fundamentally different from ours, so we did not include them as direct baselines. While one could in principle train such an approach on the union of source and target data and compare it in our setting, under this formulation it becomes conceptually closer to standard offline RL, for which our IQL based baseline is more appropriate. For this reason, we chose to report IQL* as the primary comparison in our experiments. Instead, we have included these approaches in the related work section of the revised manuscript.
>
> **Question 4 (Reward and inverse model setup):**
>
> Regarding the inverse model, since it simply infers the action that produces $s'$ from a given state $s$ by modeling $P(a \mid s, s')$, it can be trained in essentially the same way as a standard dynamics model $P(s' \mid s, a)$, as in prior work [R.4,R.5]. The reward model is also trained in a standard manner: it does not generate rewards from $(s, s')$ alone, but instead takes the action $\hat{a}$ predicted by the inverse model and learns the reward from $(s, \hat{a}, s')$, which is fully consistent with conventional reward modeling setups.
>
>
> [R.1] Shin et al. "Strainer GAN: Filtering out Impurity Samples in GAN Training." SIGGRAPH Asia 2024 Posters. 2024. 1-2.
>
> [R.2] Cetin et al. "Domain-robust visual imitation learning with mutual information constraints." arXiv preprint arXiv:2103.05079 (2021).
>
> [R.3] Choi et al. "Domain adaptive imitation learning with visual observation." NIPS (2023): 44067-44104.
>
> [R.4] Paster et al. "Planning from pixels using inverse dynamics models." arXiv preprint arXiv:2012.02419 (2020).
>
> [R.5] Brandfonbrener et al. "Inverse dynamics pretraining learns good representations for multitask imitation." NIPS (2023): 66953-66978.

---

> > ### Comment · Reviewer_fFKD · 2025-11-27
> >
> > Thank you for the authors’ response. Most of my concerns have been addressed. However, I am still not fully convinced about the distinction between cross-domain offline RL and meta offline RL.
> >
> > One final question: From my understanding, cross-domain offline RL appears to be one specific case of offline meta RL. In offline meta RL methods such as MetaDT or PromptDT, the problem definition explicitly involves training on tasks with different dynamics and reward functions. Doesn't this imply that meta offline RL is inherently capable of handling cross-domain problems? If so, why not simply use meta offline RL methods, which typically exhibit stronger generalization capabilities compared to cross-domain offline RL approaches?

---

> > > ### Author Response · Authors · 2025-11-29
> > >
> > > We sincerely thank the reviewer for the insightful comments. We would first like to emphasize that, although both the cross-domain offline setup and the meta offline setup learn from offline datasets collected across multiple tasks, their learning objectives are fundamentally different. For example, meta offline RL typically learns task contexts from the offline data and then conditions policies on these contexts so that the agent can generalize to slightly different tasks within a similar domain. It does not, however, explicitly aim to leverage data from one domain to assist policy learning in a completely different domain. In contrast, cross-domain offline RL assumes that the source and target domains can be significantly different, and the goal is not to learn separate policies for the source and target tasks, but rather to explicitly exploit the offline data collected in the source task in order to improve learning of the target task.
> > >
> > > To illustrate this distinction more concretely, we additionally train Meta-DT on the union of the source and target offline datasets and report this result as Meta-DT* in Table D.3 in Appendix D.3. The results show that Meta-DT* performs substantially worse than TCE and even worse than IQL*, which directly uses both datasets to optimize the target policy. This suggests that meta offline training mainly learns to solve the two domains as separate tasks and, when the domain gap between source and target is large, its generalization alone is insufficient to make the source data truly helpful for solving the target task unless one explicitly extracts and reuses source samples in a cross-domain manner. For these reasons, we believe that cross-domain offline RL requires separate consideration beyond the standard meta offline RL setup, and we hope this clarification helps address the reviewer’s concern.

---

### Official Review · Reviewer_XXfa · 2025-11-04

**Soundness:** 3
**Presentation:** 2
**Contribution:** 2
**Rating:** 2
**Confidence:** 4

**Summary:**

This paper presents two-stage coverage expansion (TCE) for cross-domain offline RL via score-based generative modeling. TCE trains a mixture-based state score network to expand the state space, and a target-transition score network to align the dynamics. Then the generated samples are filtered based on Z-score and mixed with the target dataset for data augmentation. Experiments on MuJoCo-like environments with dynamics shifts indicate TCE outperforms previous baselines.

**Strengths:**

- The paper is easy to understand and follow.
- TCE seems effective on cross-domain offline setting compared with previous baselines.

**Weaknesses:**

- The novelty is limited. It seems that TCE merely utilizes the generation capabilities of the score-based generative model in cross-domain offline RL setting, without more in-depth analysis or investigations. Therefore, the inspiration of this work is limited. I think an ICLR paper should include more inspirations and analysis.

- The source dataset is not fully utilized. It seems that only the state score network training uses the states in source dataset, while other information in the source dataset such as source dynamics is disgarded.

- The reward model and inverse dynamics model are trained on target datasets. Then how can they be reliable for the generated samples which may contain OOD data? No more analysis or error bound are provided.

- The figures in this paper are blurry. Please provide vector graphics (such as pdf, svg) instead of png or jpg.

**Questions:**

Please see the weaknesses for the concerns. I think this paper lacks of novelty and depth for an ICLR paper and recommend for rejection.

---

> ### Author Response · Authors · 2025-11-22
>
> **Weakness 1 (Novelty issue):**
>
> We sincerely appreciate the reviewer’s insightful comments regarding the novelty of our work. Our proposed TCE method is explicitly motivated by the limitations of existing distance based approaches that directly reuse source data. In conventional methods, a subset of source data is selected purely based on distance in the cross domain setting, and only this filtered subset is used for training. As we show in our motivation experiments, when the discrepancy between the source and target domains is large, such distance based selection can in fact significantly deteriorate the learning performance.
>
> Following the reviewer’s suggestion, we have strengthened the theoretical and empirical analysis of TCE. In particular, **Theorem 1 provides a gap bound analysis that highlights the necessity of TCE by showing that a mixture of source data and TCE generated samples can achieve a tighter performance gap bound than using only source data.** Motivated by this result, we additionally compare (i) TCE (OG), which uses only generated samples, and (ii) TCE (NN), which combines a distance based source selection, in the spirit of prior work, with TCE generated samples. These ablations, together with a discussion on the reliability of generated samples, offer a more in depth understanding of how TCE improves performance and clarify its distinct role beyond traditional distance based methods. We believe these additions substantially enhance both the quality and the novelty of the paper, and we are grateful to the reviewer for encouraging us to make these points more explicit.
>
> **Weakness 2 (Use of source data):**
>
> We thank the reviewer for raising the important question of how source data should be used in conjunction with TCE. In the original submission, we mainly demonstrated that using only samples generated by TCE already leads to significant performance improvements, but we did not thoroughly analyze when and how the original source data can further contribute. As stated in the above answer, we address this point more explicitly by separating the analysis into TCE (OG) and TCE (NN). Theoretically, Theorem 1 suggests that using source data is not always beneficial. When the discrepancy between source and target is too large, that is when $TV(P_{\text{src}} \parallel P_{\text{tar}})$ is large, relying solely on TCE generated samples is preferable, since the source distribution can introduce harmful bias. On the other hand, if the target data quality is low or the target dataset is too narrow so that overfitting becomes severe, then, as the reviewer pointed out, appropriately using source data can be advantageous.
>
> Empirically, our experiments reflect this behavior: in Table 1 and Table 2, where morphology and kinematic are substantially changed, TCE (OG) typically outperforms variants that use source data, which shows that excluding source data and using only TCE samples is beneficial in such cases. In contrast, Table D.1 shows environments where TCE (NN) performs better, indicating that source data can help when the domain gap is moderate or the target data are limited. These comparisons clearly delineate scenarios in which source data should or should not be used, and we believe this directly addresses the reviewer’s concern.
>
> **Weakness 3 (Reliability of reward and inverse model):**
>
> We appreciate the reviewer’s careful observation that our method relies on the reliability of generated samples, which is consistent with the assumption in Theorem 1 that the approximate dynamics $\hat P_{\mathrm{tar}}$ should be close to the true dynamics $P_{\mathrm{tar}}$. To assess the reliability of two-stage approach, we analyze the reliability of TCE transition generation by measuring the distributional discrepancy between generated and true target transitions as a function of $y_{\max}$, where $y_{\max}$ is a coverage control parameter, as illustrated in Fig. 3(a). From the result, we observe in Fig. 3(b) that when $y_{\max} \le 0.2$ the KL divergence remains small, indicating that the generated transitions are reasonably faithful to the target dynamics in this regime.
>
> In addition, we agree that further consideration is needed for the reward and inverse models. For the inverse model, a direct comparison is difficult because there is no explicit target policy that maps states to actions. For the reward model, however, we explicitly compare the rewards predicted for generated samples with those from the true reward function, as reported in Fig. B.3. **This reward analysis confirms that up to $y_{\max} = 0.2$ the errors with respect to real samples remain small, which supports the conclusion that the generated samples used by TCE are sufficiently reliable in the regime where we apply the method.**
>
> **Weakness 4 (Figure resolution):**
>
> We thank the reviewer for this helpful suggestion and have improved the visibility of all figures by replacing them with higher resolution versions in the PDF.

---

### Author Response · Authors · 2025-11-22

We sincerely thank all reviewers for their constructive feedback. Following your suggestions, we have substantially strengthened the manuscript with additional theoretical analyses and experiments to better demonstrate the necessity, reliability, and practical advantage of our framework. A revised version, with all changes highlighted in blue, has been uploaded. The major updates are summarized below.

**(i) Theoretical foundation of TCE (Section 4, Theorem 1, Appendix H):**

To address concerns regarding theoretical justification, **we added Theorem 1, which provides a rigorous gap bound analysis for the mixture of source data and generated transitions.** This analysis explicitly justifies our design by characterizing the trade-off between source data selection;TCE(NN) and pure generative expansion;TCE(OG), clarifying when each strategy effectively reduces the performance gap.

**(ii) Enhanced reliability analysis and Z-score validation (Figure 3(b), Appendix F):**

Beyond the initial transition analysis in Fig. 3(b), we have substantially deepened the reliability study for both generated states and rewards.
Using NN-distance analysis, we first verified that the generated states lie sufficiently close to the true data. Crucially, we further conducted a separate error analysis on the outlier samples rejected by Z-score filtering. As shown in the revised Fig. 3(b), these outliers consistently exhibit significantly higher transition and reward errors compared to the retained samples. This quantitative evidence confirms that Z-score filtering effectively identifies and discards low-fidelity samples that contradict learned target dynamics. Appendix F demonstrates that these results are consistent across diverse environments. We hope that these results firmly validate our method’s reliability and the effectiveness of Z-score filtering.

**(iii) Additional performance comparisons and comprehensive ablation studies (Section 5, Appendix D, E, G):**

We expanded our analysis to rigorously validate TCE. First, we benchmarked TCE against a recent baseline (Appendix D.3), achieving highly competitive performance. We also extended evaluations to extreme domain gaps (e.g., gravity shifts in Appendix D.2), where TCE demonstrates superior robustness over standard baselines. Methodologically, Table 3 and Appendix G confirm the contribution of each component: Appendix G.1 shows stable performance across varying denoising steps, while Appendix G.2 validates the necessity of our two-stage framework over simple target-style augmentation. Appendix E reports computational costs across diverse environments, confirming reasonable training overhead despite significant variations in state dimensions.

**For some reviewers, due to the space limit, we have split our response into two parts.** We believe these revisions and responses address the main concerns raised during review and improve the clarity and completeness of the paper. We are grateful for the reviewers’ guidance, which materially enhanced the manuscript.

---

### Comment · Area_Chair_htvD · 2025-11-26

Dear Reviewers,

Thank you for your time and effort in reviewing the submission. A reminder that the author–reviewer discussion period is about to conclude in one week. If you have not already done so, please review the authors’ rebuttals and engage in the discussion with the authors. Thanks!

Best,
Your AC

---

### Author Response · Authors · 2025-12-02

Dear Area Chair,

Thank you very much for overseeing our submission. We greatly appreciate your time and effort in handling our paper. In addition to our detailed point-by-point responses and common responses to the reviewers, we summarize below what we regard as the main contribution of our work and how the rebuttal process led us to refine and improve the manuscript, in the hope that this will assist your assessment.
___
**Main contribution:** The proposed TCE method tackles a critical problem in cross-domain offline RL: existing approaches typically focus only on selecting a subset of offline source data for training, which we show can significantly hinder target-domain learning, since even carefully distance-based selection of source data can yield worse performance than using the limited target data alone. To address this, we introduce a theorem showing that **mixing source data with target-like generated transitions can yield a tighter gap bound** in the target domain, and instantiate this idea using two generative models, one that produces mixture states from source and target data and another that generates target-like transitions from these states, together with outlier filtering and regularization to control sample quality and avoid overfitting under scarce target data. Finally, **guided by our theoretical analysis on the mixture weight** $\lambda$ **between source and generated samples, we define two variants, TCE (OG), which uses only generated samples, and TCE (NN), which additionally leverages source samples**, allowing TCE to adapt to environments where source data are helpful or harmful; across nearly all environments these variants substantially outperform existing algorithms and in some cases succeed where prior methods essentially fail, demonstrating that TCE provides a significant contribution to cross-domain offline RL.
___
**Summary of rebuttal:** As a result of the rebuttal process, the reviewers largely agreed on the necessity of the proposed approach for cross-domain offline RL and on the strength of our empirical evaluation. The remaining concerns focused primarily on the theoretical foundation of TCE (reviewers *XXfa*, *hkUn*), the reliability of the generated samples and the effectiveness of our filtering strategy (all reviewers), and the need for broader experiments and analyses (reviewers *z9Ke*, *RPuV*, *fFKD*). In the revision, we introduced **Theorem 1 to provide a formal gap bound analysis for TCE and clarified how different mixture strategies affect this bound**, explicitly aligning the theory with the two practical variants TCE (OG) and TCE (NN); we also **added detailed analyses of sample reliability and Z-score filtering**, including comparisons between generated and real states and to a target model trained with abundant target data, and extended the empirical study with environments exhibiting more extreme domain gaps, additional baselines, and more fine-grained hyperparameter analyses. We believe these changes address the key concerns raised in the reviews and substantially strengthen both the clarity and the technical soundness of the paper.
___
Due to an OpenReview system issue, it is unfortunate that we were not able to receive follow-up comments from most reviewers. Reviewer *fFKD* explicitly stated that most of their concerns had been resolved, and their remaining issue concerned comparison with meta-DT. In response, we conducted additional experiments and clarified the conceptual and empirical differences between TCE and meta-offline RL methods. Although the other reviewers could not provide follow-up remarks, their concerns were very similar to those of *fFKD* (theoretical justification and sample reliability), so we believe that our revised paper and response would likely have resolved most of their concerns as well.

In summary, we believe that TCE provides a distinct and meaningful contribution to cross-domain offline RL, and that the manuscript has been notably strengthened through the rebuttal process. The main concerns in the initial reviews largely focused on a common set of points, which we have carefully addressed by refining the theoretical analysis, clarifying the link between the theory and the practical algorithm, and adding detailed empirical studies on sample reliability and filtering. We hope these revisions clarify our contribution and support your final decision, and we are sincerely grateful for your time and thoughtful consideration of our submission.

---

### Meta-Review · Area_Chair_xYja · 2026-01-10

**Summary:**

This paper aims at resolving the distribution mismatch in resuing source transitions. Although there are some merits, most reviewers (4/5) raised plenty of concerns and reached a consensus. Therefore, I recommend it for rejection.

**Reviewer Scores:**

No, the reviewers may insist the initial ratings.

---

### Decision · Program_Chairs · 2026-01-26

Reject